# A new daily gridded precipitation dataset for the Chinese mainland based on gauge observations

Jingya Han, Chiyuan Miao[*], Jiaojiao Gou, Haiyan Zheng, Qi Zhang, Xiaoying Guo

State Key Laboratory of Earth Surface Processes and Resource Ecology, Faculty of Geographical Science, Beijing Normal University, Beijing 100875, China

\* Corresponding author: Chiyuan Miao (miaocy@bnu.edu.cn)

## Abstract

High-quality, freely accessible, long-term precipitation estimates with fine spatiotemporal resolution play essential roles in hydrologic, climatic, and numerical modeling applications. However, the existing daily gridded precipitation datasets over China either are constructed with insufficient gauge observations or neglect topographic effects and boundary effects on interpolation. Using daily observations from 2,839 gauges located across China and nearby regions from 1961 to the present, this study compared eight different interpolation schemes that adjusted the climatology based on a monthly precipitation constraint and topographic characteristic correction, using an algorithm that combined the daily climatology field with a precipitation ratio field. Results from these eight interpolation schemes were validated using 45,992 high-density daily gauge observations from 2015 to 2019 across China. Of these eight schemes, the one with the best performance merges the Parameter-elevation Regression on Independent Slopes Model (PRISM) in the daily climatology field and interpolates station observations into the ratio field using an inverse distance weighting method. This scheme had median values of 0.78 for the correlation coefficient, 8.8 mm/d for the root-mean-square deviation, and 0.69 for the Kling-Gupta efficiency for comparisons between the 45,992 high-density gauge observations and the best interpolation scheme for the 0.1° latitude × longitude grid cells from 2015 to 2019. This scheme had the best overall performance, as it fully considers topographic effects in the daily climatology

field and it balances local data fidelity and global fitting smoothness in the interpolation of the precipitation ratio field. Therefore, this scheme was used to construct a new long-term, gauge-based gridded precipitation dataset for the Chinese mainland (called CHM_PRE, as a member of the China Hydro-Meteorology dataset) with spatial resolutions of 0.5°, 0.25°, and 0.1° from 1961 to the present. This precipitation dataset is expected to facilitate the advancement of drought monitoring, flood forecasting, and hydrological modeling. Free access to the dataset can be found at https://doi.org/10.6084/m9.figshare.21432123.v4 (Han and Miao, 2022).

## 1 Introduction

As one of the key components of the hydrological cycle, precipitation can influence the distribution of water resources (Rodell et al., 2018), sustain agriculture (Beck et al., 2020; Zou et al., 2022), replenish aquifers (Fischer and Knutti, 2016; Kucera et al., 2013), and enable economic prosperity (Trenberth et al., 2003; Kirschbaum et al., 2017). Each of the last three decades has been successively warmer at the Earth's surface than any preceding decade since 1850 (IPCC, 2021). With an ever-warming climate, the Earth's water cycle has been amplified, resulting in frequent severe extreme precipitation events (Fischer and Knutti, 2016; Myhre et al., 2018). The intensification of water transport and exchanges between the atmosphere and land surface is having profound impacts on the redistribution of water resources by moisture flux, which exaggerates the contrast between wet and dry meteorological regimes, seasons, and events (Allen and Ingram, 2002; Allan et al., 2020; Han et al., 2021). Constructing a high-quality, long-term daily precipitation dataset is essential for hydrometeorological research (Sun et al., 2018; Beck et al., 2019). However, due to the spatial heterogeneity and temporal variability of daily precipitation, it is challenging to derive accurate spatiotemporal patterns of daily precipitation.

Collection of precipitation data relies mainly on measurements using ground-based rain gauges, and estimates using remote sensing technologies such as weather radar and satellite (Shen et al., 2014; Beck et al., 2019; Sun et al., 2018). Among these approaches,

rain-gauge observations are the most reliable and widely used tool for directly measuring precipitation. However, precipitation data measured with gauges are point observations only, and the uneven distribution of gauges increases the limitations of gauge applications over a region. It is vital to interpolate these spatially irregular gauge observations to areal averages, since multiple scientific and operational applications (e.g., estimating local climate variables in data-sparse regions, monitoring climate change at the regional or global scale, and validating climate models with observations) require good-quality, high-spatiotemporal-resolution precipitation datasets (Haylock et al., 2008; Xie et al., 2007; Harris et al., 2020). Spatial interpolation methods are usually applied to irregular point observations to produce an evenly distributed precipitation grid for application in hydrological and meteorological studies (Ahrens, 2006; Schamm et al., 2014; Golian et al., 2019).

In China, the original gauge observations used as the benchmark of various precipitation datasets mainly come from two suites of gauge observations provided by the China Meteorological Administration (CMA): ~700 benchmark stations and ~2,400 gauges comprising other ordinary national automatic weather stations (Shen and Xiong, 2016). Using observations from the former (the ~700 stations), a monthly precipitation dataset has been established over China that covers the period of 1901–2017 (Peng et al., 2019). To achieve a higher temporal resolution, a daily gridded precipitation dataset has been produced for China using the same raw precipitation data for the same period, from 1961 to 2019 (Qin et al., 2022). Through a fusion of remote sensing products, reanalysis datasets, and *in-situ* station data, the China Meteorological Forcing Dataset (CMFD) has been produced to serve as a high-resolution (three hours, 0.1° × 0.1°) input forcing dataset for hydrological and ecosystem models beginning in 1979 (He et al., 2020). Generally, the quantitative accuracy of a gauge-based dataset can be improved by enhancing the density of gauge observations (Merino et al., 2021; Hofstra and New, 2009). Xie et al. (2007) developed a widely used gauge-based analysis of daily precipitation over East Asia (EA05) with a collection of daily precipitation observations from over 700 stations from CMA and about 1,000 hydrological station observations

from the Chinese Yellow River Conservation Commission. Using observations from approximately 2,400 gauges from 1961 to the present, Wu and Gao (2013) created a daily gridded dataset with a resolution of 0.25° × 0.25° over China (CN05.1), and Zhao et al. (2014) constructed the second version of a 0.5° × 0.5° gridded daily precipitation dataset over China (CMA V2.0). Further accounting for topographic effects, including elevation, slope, proximity to coastlines, and the locations of temperature inversions, Shen et al. (2010) developed the China Gauge-based Daily Precipitation Analysis (CGDPA) with spatial resolutions of 0.5° × 0.5° and 0.25° × 0.25° using a topographic correction algorithm. The aforementioned datasets only involve gauges inside China's boundaries, except for EA05, which covers the East Asia domain. This limitation can lead to boundary effects such that grid cells near the boundaries suffer positioning inaccuracy in relation to interior grid cells (Ahrens, 2006). In addition, different interpolation algorithms can produce different results even with the same inputs. Comparing the performance of different interpolation techniques is crucial to determining the best interpolation method. A summary of these gridded precipitation datasets is shown in Table 1.

Table 1 Gauge-based gridded precipitation datasets for China

| Name | Spatial resolution | Domain | Temporal resolution | Time period | Reference | Number of stations | Interpolation method |
|---|---|---|---|---|---|---|---|
| 1 km monthly temperature and precipitation dataset for China from 1901 to 2017 | 1 km | China | Monthly | 1901 to the present | Peng et al., 2019 | ~700 | Bilinear interpolation |
| HRLT | 1 km | China | Daily | 1961–2019 | Qin et al., 2022 | ~700 | Machine learning, the generalized additive model, and thin plate spline |
| CMFD | 0.1° × 0.1° | China | Three hours | 1979 to the present | He et al.,2020 | ~700 | Thin plate spline |
| EA05 | 0.5° × 0.5° | East Asia | Daily | 1978–2003 | Xie et al., 2007 | ~1,700 | Optimal interpolation |
| CN05.1 | 0.25° × 0.25° | China | Daily | 1961 to the present | Wu and Gao, 2013 | ~2,400 | Angular distance weight |
| CMA V2.0 | 0.5° × 0.5° | China | Daily | 1961–2019 | Zhao et al., 2014 | ~2,400 | Thin plate spline |
| CGDPA | 0.25° × 0.25°, 0.5° × 0.5° | China | Daily | 2008–2015 | Shen et al., 2010 | ~2,400 | Optimal interpolation |

Given these limitations and the important role these datasets play in many applications, it is of great urgency to establish long-term, continuously updated daily precipitation series with multiple spatial resolutions that are free to use. Therefore, this study aimed to construct a long-term (from 1961 to the present) daily precipitation dataset with different spatial resolutions ($0.5° \times 0.5°$, $0.25° \times 0.25°$, and $0.1° \times 0.1°$) based on 2,839 gauge observations in and around China (2,419 gauges located across the Chinese mainland and 420 gauges from nearby regions). Eight interpolation schemes were considered and validated using 45,992 gauge observations for the period of 2015–2019 over China. Finally, we produced a new gridded precipitation dataset for the Chinese mainland (a member of the China Hydro-Meteorology datasets, hereinafter called CHM_PRE) covering the period 1961–2022, with spatial resolutions of 0.5°, 0.25°, and 0.1°, which is available in the public domain and will be updated yearly.

## 2 Data

### 2.1 Raw gauge data used for interpolation

Daily rain gauge datasets (from 1961 to the present) from 2,419 stations across the Chinese mainland and 420 stations just outside China's boundaries were collected from the China Meteorological Administration (CMA, http://data.cma.cn) and Global Historical Climatology Network-Daily Version 3 (GHCND, https://www.ncei.noaa.gov), respectively (Figure 1a). The CMA gauge dataset is available for the Chinese mainland (data in Hong Kong, Macao, and Taiwan are currently not accessible for use). Stations are sparsely distributed in northwestern China and the Tibetan Plateau compared with eastern and southern China. The daily precipitation is the accumulated precipitation amount between 20:00 and 20:00 (local time in Beijing). This dataset has been subjected to strict quality controls, including (1) an extreme values check, (2) an internal consistency check of daily values, (3) a spatial and temporal consistency check, and (4) manual verification (Zhang et al., 2020). The gauge observations from neighboring countries come from the GHCND dataset, which contains records from over 80,000 stations in 180 countries and territories. Quality controls are routinely applied to assure the basic consistency of the dataset (Menne et

al., 2012). Stations with less than 5% of calendar days missing in an individual year were used for interpolation. Changes in the number of stations over time are shown in Figure 1c.

**2.2 High-density gauge observations used for validation**

High-density daily observations from nearly 68,000 automatic weather stations for the period 2015–2019 in China are provided by the National Meteorological Information Center of CMA (Li et al., 2018). Once stations with more than 20% missing data were removed, there were 45,992 good-quality stations available for validation (Figure 1b).

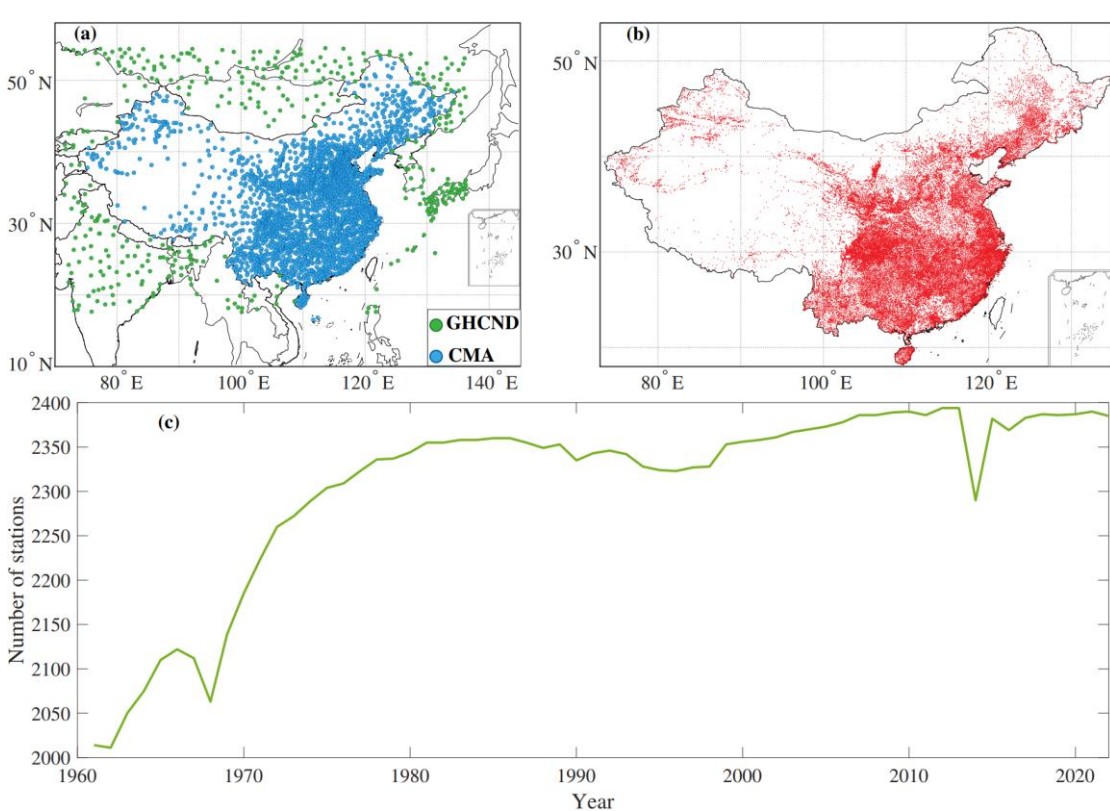

Figure 1. (a) Distribution of 2,839 stations used in interpolation and (b) 45,992 stations used for validation. (c) Quality-controlled number of stations for interpolation over time.

**2.3 SRTM-DEM**

The 3-arc second (90 m resolution) digital elevation model (DEM) applied in this study was acquired from the Shuttle Radar Topography Mission (SRTM) data (https://cmr.earthdata.nasa.gov/search/concepts/C1214622194-SCIOPS). SRTM uses

dual radar antennas to acquire interferometric radar data and process digital topographic data (Farr et al., 2007). We resampled the SRTM-DEM into 0.05° × 0.05° grid cells using the bilinear interpolation method.

**2.4 PRISM**

The monthly climatology generated by the Parameter-elevation Regression on Independent Slopes Model (PRISM) was used for the monthly precipitation constraint and topographic characteristic correction of the daily climatology field (https://prism.oregonstate.edu/). PRISM incorporates local climate-elevation relationships, topographic features, proximity to coastlines, and several measures of terrain complexity, and it is the most widely used climatology dataset in the world (Daly et al., 1994; Daly et al., 2002). The original spatial resolution is 0.04° × 0.04° for the monthly climatology of PRISM between 1961 and 1990; we used bilinear interpolation to regrid the spatial resolution into 0.05° × 0.05° grid cells for adjustment based on climatology.

**3 Methodology**

**3.1 Interpolation scheme**

Due to the high spatial variability of precipitation relative to other climate variables, directly interpolating the daily rain-gauge observations into grid cells could produce a dataset with misleading daily precipitation characteristics (Xie et al., 2007; Chen et al., 2002; Shen et al., 2010). To avoid this and reduce introduced errors, the overall strategy for establishing a daily gridded precipitation dataset is to construct a relatively continuous daily climatology field (Shen et al., 2010). Then, we would build an intermediate field of the interpolated variable based on this daily climatology field, such as a daily precipitation anomalies field or a field of the ratio between daily precipitation and daily climatology. Previous studies have demonstrated that interpolating the ratio (between daily precipitation and daily climatology) yields better performance than interpolating anomalies for constructing daily gridded precipitation (Xie et al., 2007; Yatagai et al., 2012; Di Luzio et al., 2008). Therefore, the "daily climatology field ($Cd$)

× field of the ratio between daily precipitation and daily climatology (*P/Cd*)" was
employed as the interpolation scheme for constructing the new gridded precipitation
dataset in this study (Figure 2), as developed by Xie et al. (2007). Figure 3 shows a
flowchart of the gridding analysis system.

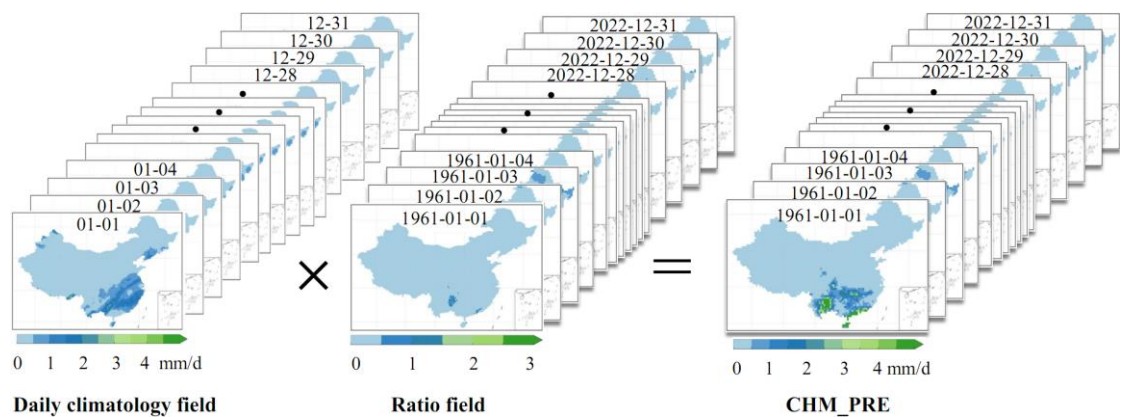

Figure 2. Interpolation strategy for generating the daily gridded precipitation dataset
(CHM_PRE) in this study.

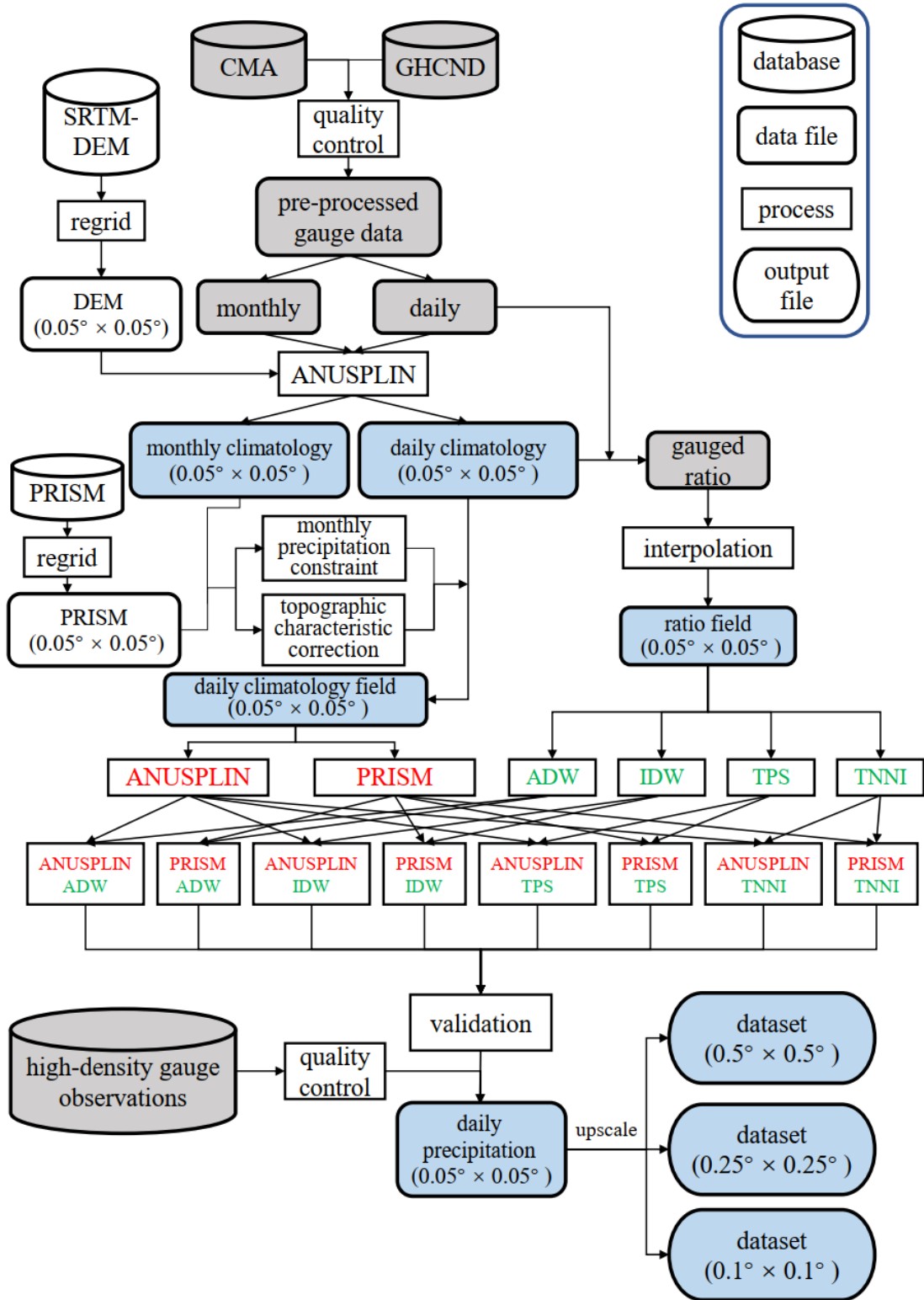

Figure 3. Flowchart of the gridding analysis system. Gray shading represents gauge data; blue shading represents gridded data. Approaches used for generating daily climatology fields (using ANUSPLIN software or PRISM data) are marked in red, and interpolation methods used for producing ratio fields are marked in green.

### 3.2 Building the daily climatology field ($Cd$)

First, the gauge-based climatology of daily precipitation was calculated using gauge observations. The definition of gauge-based climatology of daily precipitation is the Fourier-truncated 30-year mean daily precipitation series produced from gauge observations for the period of 1971–2000 for each of the 365 calendar days (Figure 4). We used Fourier truncation to remove the high-frequency noise of the 30-year mean daily precipitation series for each station and retained the accumulation of the first six harmonic components as the gauge-based climatology of daily precipitation (Xie et al., 2007). After Fourier truncation, approximately 75% of all stations preserve a variation of 40% to 75% in the truncated mean daily precipitation series relative to the total variation in the mean daily precipitation. The unadjusted 0.05° × 0.05° gridded daily climatology field ($Cd_0$) was then interpolated from the gauge-based climatology of daily precipitation with SRTM-DEM as a covariate using the ANUSPLIN software (Hutchinson and Xu, 2004). To minimize systematic bias from the unadjusted 0.05° × 0.05° gridded daily climatology field on the monthly climatology field ($Cm$), the monthly accumulation of the unadjusted 0.05° × 0.05° gridded daily climatology field was then constrained by the monthly climatology field. This produced an adjusted gridded daily climatology field that uses a monthly precipitation constraint and topographic characteristic correction. We compared two types of gridded monthly climatology fields to determine which adjusts better for the systematic bias: 1) an ANUSPLIN-type gridded monthly climatology field or 2) a PRISM-type gridded monthly climatology field. The ANUSPLIN-type gridded monthly climatology field was produced by interpolating monthly precipitation climatology (1971–2000) from stations to the 0.05° latitude × longitude grids with a covariate of SRTM-DEM, using the ANUSPLIN software. The 0.05° × 0.05° regridded monthly climatology of PRISM was used as the PRISM-type monthly climatology field.

The climatology adjustment steps for one grid cell were as follows:

1) Calculate $Cd_{0\_(m,j)}$ ($m = 1, 2, 3, \ldots, 12; j = 1, 2, 3, \ldots, 365; m$ is the corresponding month for day $j$), which is the monthly total of the unadjusted $0.05° \times 0.05°$ gridded daily climatology field, derived by taking the sum of the unadjusted $0.05° \times 0.05°$ gridded daily climatology field for the month.

2) Match the monthly total series derived using the unadjusted $0.05° \times 0.05°$ gridded daily climatology field to the gridded monthly climatology field month by month.

3) Compute the scaling factor $SF_{(m,j)}$ for the individual calendar day of the unadjusted $0.05° \times 0.05°$ daily climatology field to the gridded monthly climatology field:

$$SF_{(m,j)} = \frac{C_{(m,j)}}{w_{(m-1,j)}\, Cd_{0\_(m-1,j)} + w_{(m,j)}\, Cd_{0\_(m,j)} + w_{(m+1,j)}\, Cd_{0\_(m+1,j)}} \qquad (1)$$

$$(m = 1, 2, 3, \ldots, 11, 12; \; j = 1, 2, 3, \ldots, 365;$$

$$m \text{ is the corresponding month for day } j)$$

where $C_{(m,j)}$ is the gridded monthly climatology field for the corresponding month $m$ of day $j$; $Cd_{0\_(m-1,j)}$, $Cd_{0\_(m,j)}$ and $Cd_{0\_(m+1,j)}$ are the monthly total of months $m - 1$, $m$, and $m + 1$, respectively, which are calculated from the unadjusted $0.05° \times 0.05°$

gridded daily climatology field; $w_{(m-1,j)}$, $w_{(m,j)}$, and $w_{(m+1,j)}$ are the corresponding weights for months $m - 1$, $m$, and $m + 1$, respectively, which are inversely proportional to the interval between the calendar day $j$ and the center of the month (Xie et al., 2007). Note that the weight $w_{(m-1,j)}$ is zero when $m = 1$, and so is the weight $w_{(m+1,j)}$ when $m = 12$.

4) The adjusted gridded daily climatology field ($Cd_{(m,j)}$) is defined as

$$Cd_{(m,j)} = Cd_{0\_(m,j)}\, SF_{(m,j)} \qquad (2)$$

$$(m = 1, 2, 3, \ldots, 11, 12; \; j = 1, 2, 3, \ldots, 365;$$

$$m \text{ is the corresponding month for day } j)$$

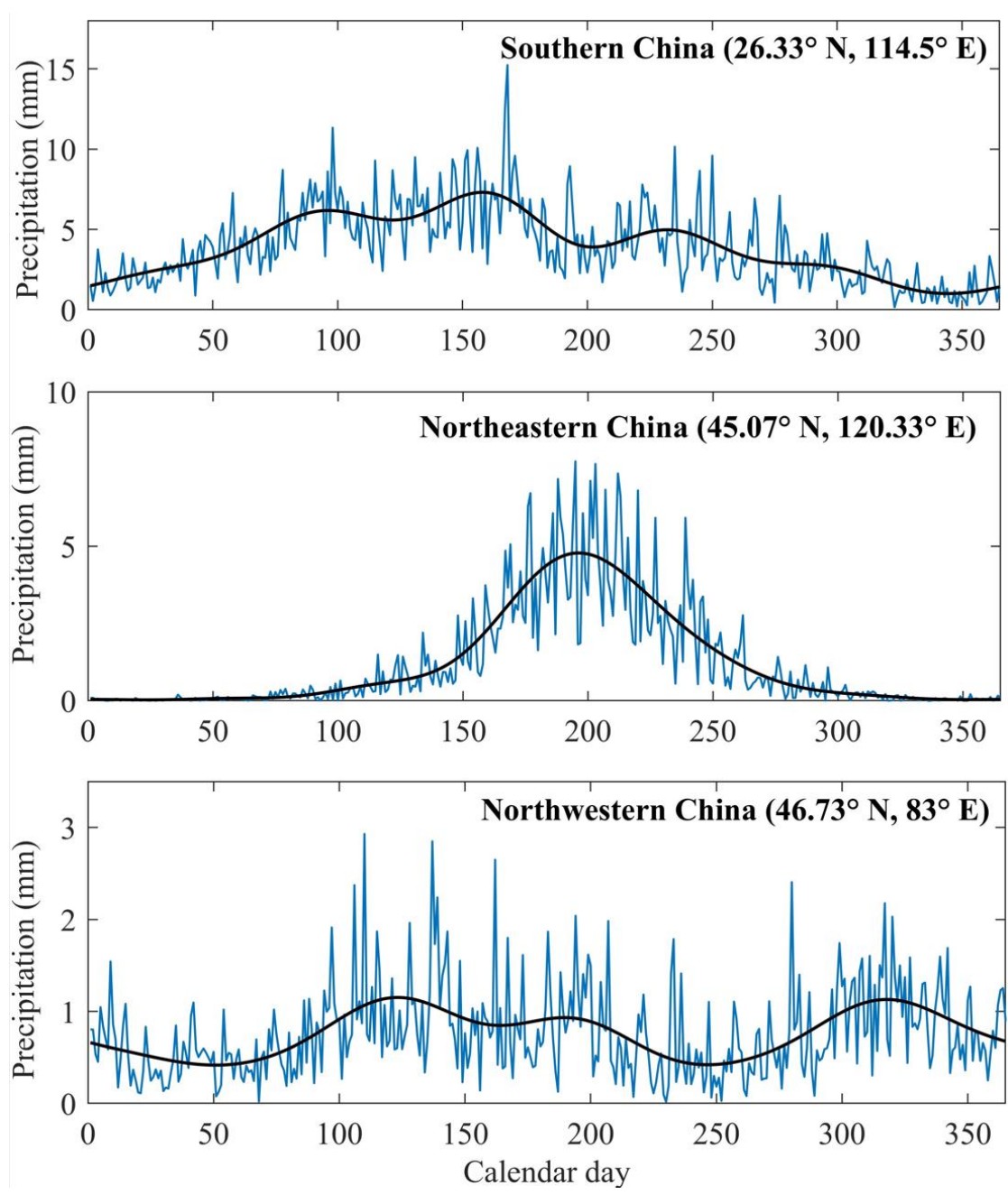


Figure 4. Time series of 30-year (1971–2000) mean daily precipitation (blue lines) and the gauge-based climatology of daily precipitation derived using Fourier truncation (black lines) for three randomly selected stations in southern China (top), northeastern China (middle), and northwestern China (bottom).


## 3.3 Constructing a field of the ratio between daily precipitation and daily climatology (*P/Cd*)

The ratio of daily precipitation to the spatiotemporally corresponding daily climatology

field was calculated for each station. To compare the performances of different

interpolation methods, four widely used interpolation approaches for precipitation were adopted. The four interpolation methods used in this study were angular-distance weighting (ADW) (Shepard, 1968; Caesar et al., 2006), inverse distance weighting (IDW) (Shepard, 1984; Eischeid et al., 2000), thin plate spline (TPS) (Hutchinson, 1995; Camera et al., 2014), and triangulation-based nearest neighbor interpolation (TNNI)

(Thiessen, 1911; Sibson, 1978). A brief overview of the main characteristics of the four methods is given below.

### 3.3.1 ADW

The ADW interpolation method used for this study was the modified Shepard's

algorithm, which introduces the concept of correlation decay distance (CDD), also called correlation length scale or decorrelation length (Shepard, 1984; Dunn et al., 2020). The CDD is defined as the distance at which the correlation between one station and all other stations decays below $1/e$, approximately corresponding to the significance level of 0.05 for the correlation within large samples (Jones et al., 1997; Harris et al.,

2020). The number of stations for interpolating the target grid cell is well constrained by the CDD, thus improving the interpolation precision (New et al., 2000; Mitchell and Jones, 2005; Hofstra and New, 2009).

For every station, correlations ($r$) and distances ($x$) with the other 2,838 stations are

shown in Figure 5, and the ordinary least squares method was used to fit an exponential decay function:

$$r = e^{-x/CDD} \tag{3}$$

The estimated CDD is 244.7 km (95% confidence interval: 244.5–244.8 km) at the 0.05 significance level.


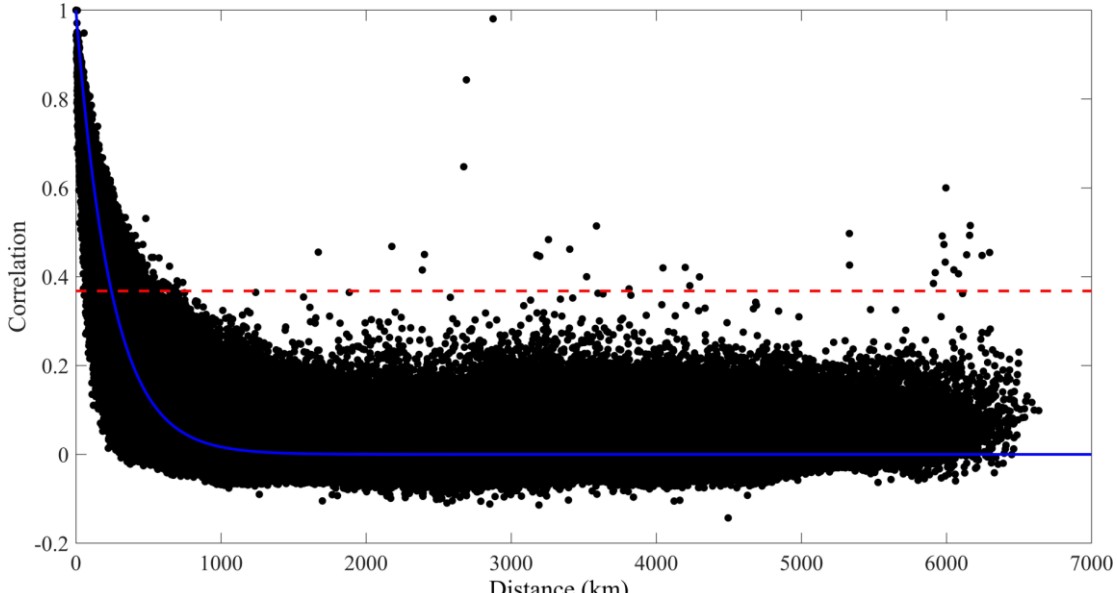

Figure 5. Estimation of correlation decay distance (CDD) for daily precipitation series for all stations in the interpolated domain. Black points show the pair of distance–correlation for each station. The blue line is the exponential curve fitted to the data by ordinary least squares. The red dashed line marks where correlation equals 1/$e$.

The ADW method accounts for the importance of both distance and the isolation of stations in interpolation (New et al., 2000). Only stations within the range of the CDD for the center of the target grid cell (L) were involved in the interpolation. The weight for each involved station ($i$) is a function of the distance weight ($D_i$) and angular weight ($A_i$):

$$D_i = \left(e^{-x_i/CDD}\right)^n \tag{4}$$

where $x_i$ is the distance between station $i$ and the center of target grid cell L; $n$ is a constant and usually set to 4, in accordance with previous studies (Harris et al., 2020; Dunn et al., 2020; Efthymiadis et al., 2006).

$$A_i = 1 + \frac{\sum_k D_k[1 - cos(\theta_k - \theta_i)]}{\sum_k D_k} \, (\, i \neq k) \tag{5}$$

where $k$ represents the surrounding stations relative to station $i$; $D_k$ is the distance weight for the surrounding stations $k$; and $\theta_i$ and $\theta_k$ are the angles relative to the north of the center of target grid cell L for station $i$ and the surrounding stations $k$. Here, "surrounding stations" refers to all other stations except for station $i$. Finally, the

weights for all contributing stations were standardized to sum to 1.0.

The angular-distance weight ($W_i$) equals

$$W_i = D_i A_i \tag{6}$$


### 3.3.2 IDW

We introduced the concept of CDD to the IDW method for interpolation. We set CDD1 = 244.7 km, which represents the boundary at which the search radius has a good correlation between stations and the target grid cell. The minimum distance satisfying the condition that at least three stations are included in the search radius is around 1,336 km. Therefore, CDD2 = 1,336 km was used as a second choice of search radius if there were not at least three stations located within the range of CDD1. We employed $S_0$ representing the target grid cell, $i$ representing the surrounding stations that fall in the search radius of the target grid cell, $y(s_i)$ denoting the station observations, and $d_{0i}$ denoting the distance between $S_0$ and $i$. The estimation of daily precipitation in grid cell $S_0$ is $\hat{y}(s_0)$:

$$\hat{y}(s_0) = \sum_{i=1}^{n} \lambda_i y(s_i) \tag{7}$$

$$\lambda_i = d_{0_i}^{-\alpha} / \sum_{i}^{n} d_{0i}^{-\alpha} \tag{8}$$

$$\sum_{i}^{n} \lambda_i = 1 \tag{9}$$

where $\lambda_i$ is the distance weight for the interpolated stations; $n$ is the number of stations involved in the interpolation; and the parameter $\alpha$ is the geometric form of weight. A high magnitude of $\alpha$ represents a strong correlation decay per unit of distance, thus the stations that are close to the target grid cell would be assigned a greater weight (Lu and Wong, 2008). In this study, we used $\alpha = 2$, which is widely used for the IDW method to show the Euclidean distance between the centers of grid cells and the interpolated stations (Ly et al., 2013; Ahrens, 2006).

### 3.3.3 TPS

Splines are developed with the use of spatial covariate functions (Wahba and

Wendelberger, 1980; Camera et al., 2014). TPS regards the spatial distribution as simply a function of observations, and there is no need to first estimate a covariate function (Hutchinson, 1995). Thus, the interpolation precision is improved. The TPS function offers a trade-off between data fidelity and smoothness of fit (Tait et al., 2006; Haylock et al., 2008). The degree of smoothing is determined by minimizing generalized cross

validation (Hutchinson, 1998).

### 3.3.4 TNNI

The Delaunay triangulation net is built by location of vertices of triangulations with rainfall amount as the third dimension (Delaunay, 1934). Within finite point sets, the

Delaunay triangulation is demonstrated to be the only optimal method (Sibson, 1978). This uniqueness guarantees the stability of interpolation. TNNI estimates the value of the target grid cell as a value of the nearest sample, and this can reflect the characteristic of regional precipitation (Vivoni Enrique et al., 2004).

### 3.4 Eight combination schemes for the daily climatology field and ratio field

The two types of daily climatology fields and four types of ratio fields constitute eight combination schemes of interpolation strategies (Table 2). We compared the performances of the eight combination schemes to choose the best scheme to construct the dataset.


Table 2 An overview of eight combination schemes for daily climatology field and ratio field

| No. | Scheme name | Interpolation method for unadjusted daily climatology field | Monthly climatology field type | Interpolation method used for ratio field |
|-----|-------------|-------------------------------------------------------------|--------------------------------|-------------------------------------------|
| 1 | ANUSPLIN + ADW | ANUSPLIN | ANUSPLIN | ADW |

| 2 | PRISM + ADW | ANUSPLIN | PRISM | ADW |
|---|---|---|---|---|
| 3 | ANUSPLIN + IDW | ANUSPLIN | ANUSPLIN | IDW |
| 4 | PRISM + IDW | ANUSPLIN | PRISM | IDW |
| 5 | ANUSPLIN + TPS | ANUSPLIN | ANUSPLIN | TPS |
| 6 | PRISM + TPS | ANUSPLIN | PRISM | TPS |
| 7 | ANUSPLIN + TNNI | ANUSPLIN | ANUSPLIN | TNNI |
| 8 | PRISM + TNNI | ANUSPLIN | PRISM | TNNI |

**3.5 Validation**

To improve the validation efficiency, 45,992 high-density gauge observations from China were used to evaluate the eight interpolation schemes with a spatial resolution of $0.1° \times 0.1°$. The validation steps are as follows:

1) Remove the 2,839 gauge stations used for interpolation from the 45,992 stations; the observations of the remaining stations are employed as the "true values".

2) Distribute the remaining stations into the corresponding $0.1° \times 0.1°$ grid cells according to their longitudes and latitudes.

3) Apply the validation only for the grid cells where stations are located. The average of station observations is calculated as the validation value for the $0.1° \times 0.1°$ grid cells where multiple stations are located. The correlation coefficient (CC), root-mean-square error (RMSE), and Kling-Gupta efficiency (KGE) between validation value ($V_n$) and estimation value ($Y_n$) are used as evaluation indicators for validation:

$$CC = \frac{\frac{1}{N}\sum_{n=1}^{N}(V_n - \bar{V})(Y_n - \bar{Y})}{\sigma_V \sigma_Y} \tag{10}$$

$$RMSE = \sqrt{\frac{1}{N}\sum_{n=1}^{N}(V_n - Y_n)^2} \tag{11}$$

$$KGE = 1 - \sqrt{(CC - 1)^2 + (\alpha - 1)^2 + (\beta - 1)^2} \tag{12}$$

$$\alpha = \frac{\sigma_Y}{\sigma_V}, \beta = \frac{\bar{Y}}{\bar{V}} \tag{13}$$

where $N$ is the length of the daily precipitation series for 2015–2019; $\sigma_V$ and $\bar{V}$ are the standard deviation and mean value for the validated daily precipitation series, respectively; $\sigma_Y$ and $\bar{Y}$ are the standard deviation and mean value for the estimated daily precipitation series, respectively; $\alpha$ is a variability bias term; and $\beta$ is a measure of

mean bias (Gupta et al., 2009).

## 4 Results and discussion

### 4.1 Best interpolation scheme derived by validation

Generally, the performances of CC, RMSE, and KGE in southeast China were all better

than those in northwest China and the Tibetan Plateau, where the density of stations is

relatively sparse (Figures 6, 7, 8). The interpolated grid values were closer to gauge

observations in simple terrain (e.g., the North China Plain) compared with complex

terrain (e.g., the Yungui Plateau, the Loess Plateau, and the Tibetan Plateau). In terms

of CC, the performances of scheme 3 (ANUSPLIN + IDW) and scheme 4 (PRISM +

IDW) were best among the eight schemes, both with median CC values of 0.78. The

median CC values for scheme 7 (ANUSPLIN + TNNI; 0.76) and scheme 8 (PRISM +

TNNI; 0.76) were a little smaller than those for schemes 3 and 4. Scheme 1

(ANUSPLIN + ADW) and scheme 2 (PRISM + ADW) shared the same median CC

value of 0.71. The lowest median CC value (0.63) was calculated for scheme 5

(ANUSPLIN + TPS) and scheme 6 (PRISM + TPS). For the median RMSE values, the

order of the eight interpolation schemes was as follows: scheme 4 (8.8 mm/d) < scheme

3 (8.83 mm/d) < scheme 1 (10.09 mm/d) = scheme 2 (10.09 mm/d) < scheme 7 (10.14

mm/d) < scheme 8 (10.23 mm/d) < scheme 5 (11.16 mm/d) < scheme 6 (11.17 mm/d).

The results for the comprehensive index, KGE, combining the characteristics of

correlation, variability bias, and mean bias show that using scheme 4 would obtain the

best interpolation results, at a median KGE value of 0.69 for China. Schemes 3, 7, and

8 performed slightly worse than scheme 4, all with the same value of 0.68. The median

KGE values for scheme 1 (0.56) and scheme 2 (0.57) were a little worse than those

listed above. Using scheme 5 and scheme 6 would result in the worst performance based

on KGE values; their median KGE values were about 0.49 and 0.5, respectively. Overall,

using scheme 4, which applies PRISM monthly climatology to adjust the daily

climatology field, combined with an IDW-interpolated ratio field, achieved the best

performance among these validation indices. Therefore, the best scheme (PRISM +

IDW) was used to construct the new 62-yr CHM_PRE dataset for the Chinese mainland.

The gridded data is the areal average precipitation over the grid cell.

Scheme 4 had better performance than the other schemes because it considers the impact of topography more deeply and holds an appropriate balance between local data fidelity and global fitting smoothness. The overall interpolation strategy was to combine

the daily climatology field ($Cd$) with the field of the ratio between daily precipitation and daily climatology ($P/Cd$). With respect to $Cd$, the PRISM-type daily climatology field incorporates topographic features, proximity to coastlines, and several measures of terrain complexity, which goes beyond the climate–elevation relationships that the ANUSPLIN-type daily climatology field considers. As for $P/Cd$, we selected the four

alternative interpolation methods (ADW, IDW, TPS, and TNNI) to consider the balance between local data fidelity and global fitting smoothness in addition to the popularity, authority, and simplicity of the interpolation methods. Specifically, the ADW and IDW methods were chosen due to their high local data fidelity. Both are local interpolation methods (Liszka, 1984). Unlike the IDW method, the ADW method assigns a tiny

weight to far-distant gauge observations to promote global fitting smoothness. This impacts the local accuracy of interpolation. The TPS and TNNI methods, on the other hand, were chosen for their high global fitting smoothness. Both TPS and TNNI are global interpolation methods (Liszka, 1984). The TPS method is based on a mathematical model for surface estimation that fits a minimum-curvature surface

through all input points, while TNNI constructs a Delaunay triangulation of three stations locations. So TNNI tends to assign more weights to maintain local data fidelity but has weaker fitting smoothness. To sum up, the combination of PRISM-type $Cd$ and IDW-type $P/Cd$ yielded the best performance among the selected schemes. This was not simply due to chance. This best-performing interpolation scheme could be applied

in other regions, but further validation would be needed to confirm whether it is the best-performing interpolation scheme there.

Due to a high rate of missing daily observations from 1961 to 1980, up to 15% of stations could not reach the threshold for quality control (i.e., a rate of missing daily

precipitation not more than 5%) and were removed. With the vigorous development of hydrometeorological observation in China since the 1980s, precipitation data quality has been improving, which caused a jump in the number of stations that met quality control requirements beginning in 1981 (Shen et al., 2014). If we had kept the number of gauges used for interpolation steady over the full 62 year-span, about 300 gauges

available for the period 1981–2022 would have been excluded, which would have been a great loss of real observed precipitation information. Therefore, the strategy adopted in this study was that all observational data that met the quality control conditions were used for data interpolation, which led to some differences in the number of sites used every year. These slight differences could be partly compensated for by using

correlation decay distance (CDD1 and CDD2), which confirmed there were at least three stations involved in the interpolation for each grid cell so that there would not be a sharp change in the number of stations used for interpolation for each grid cell.

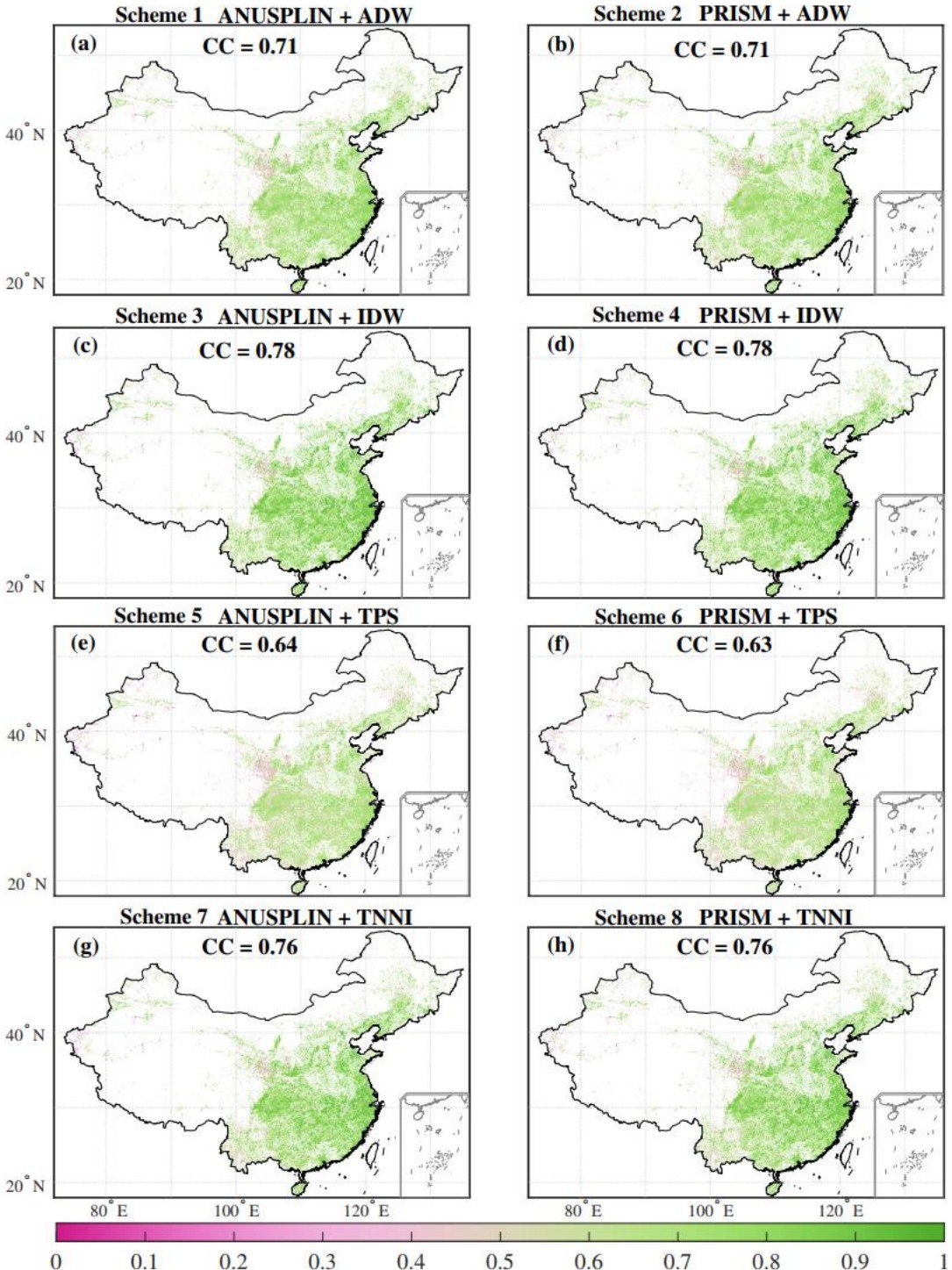

Figure 6. (a–h) Spatial pattern of the correlation coefficient (CC) for eight combination schemes. Numbers in each subplot represent the median of CC values across all grid cells involved in validation.

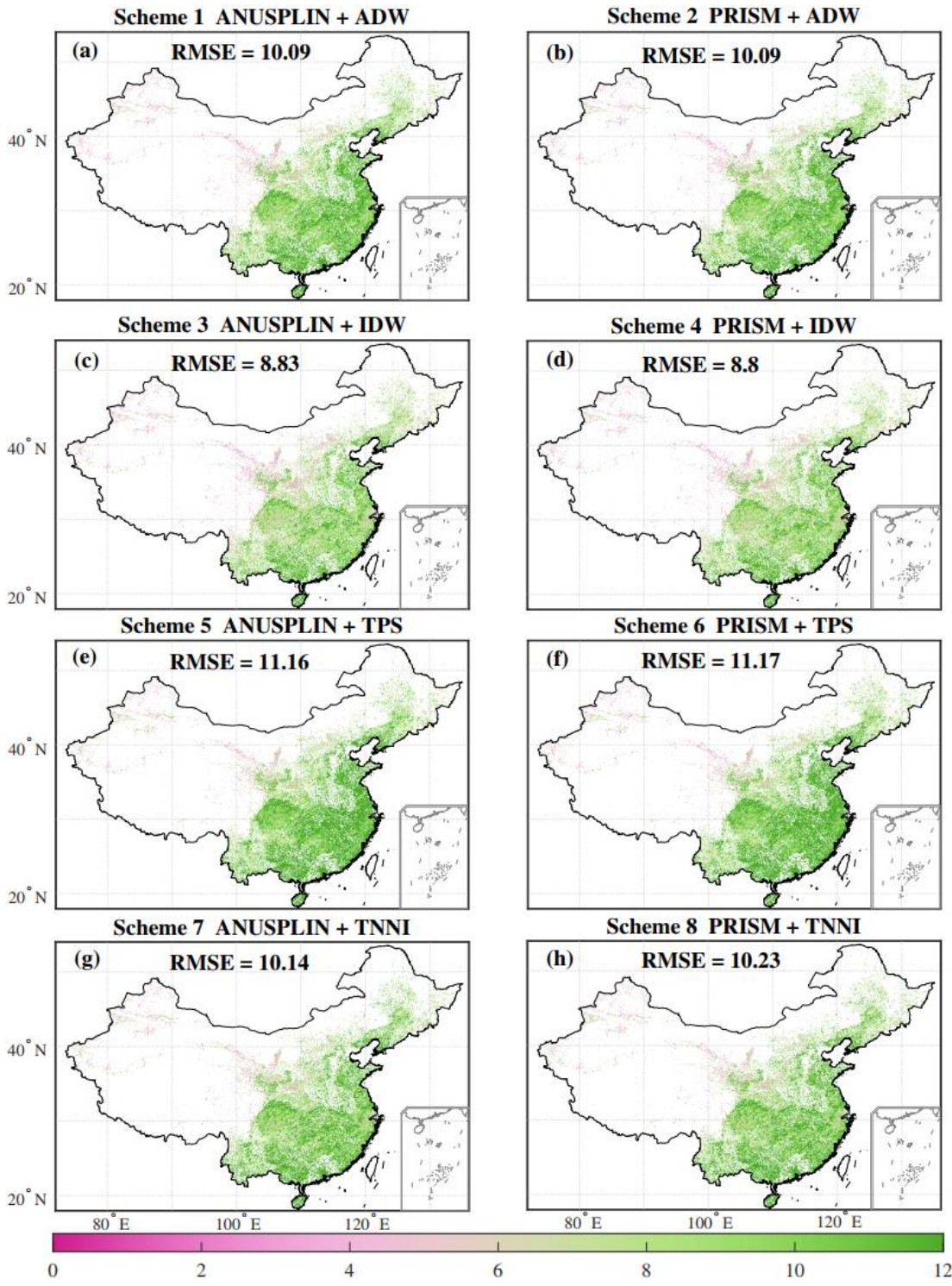

Figure 7. The same as Figure 6, but for root-mean-square error (RMSE). The unit is mm/d.

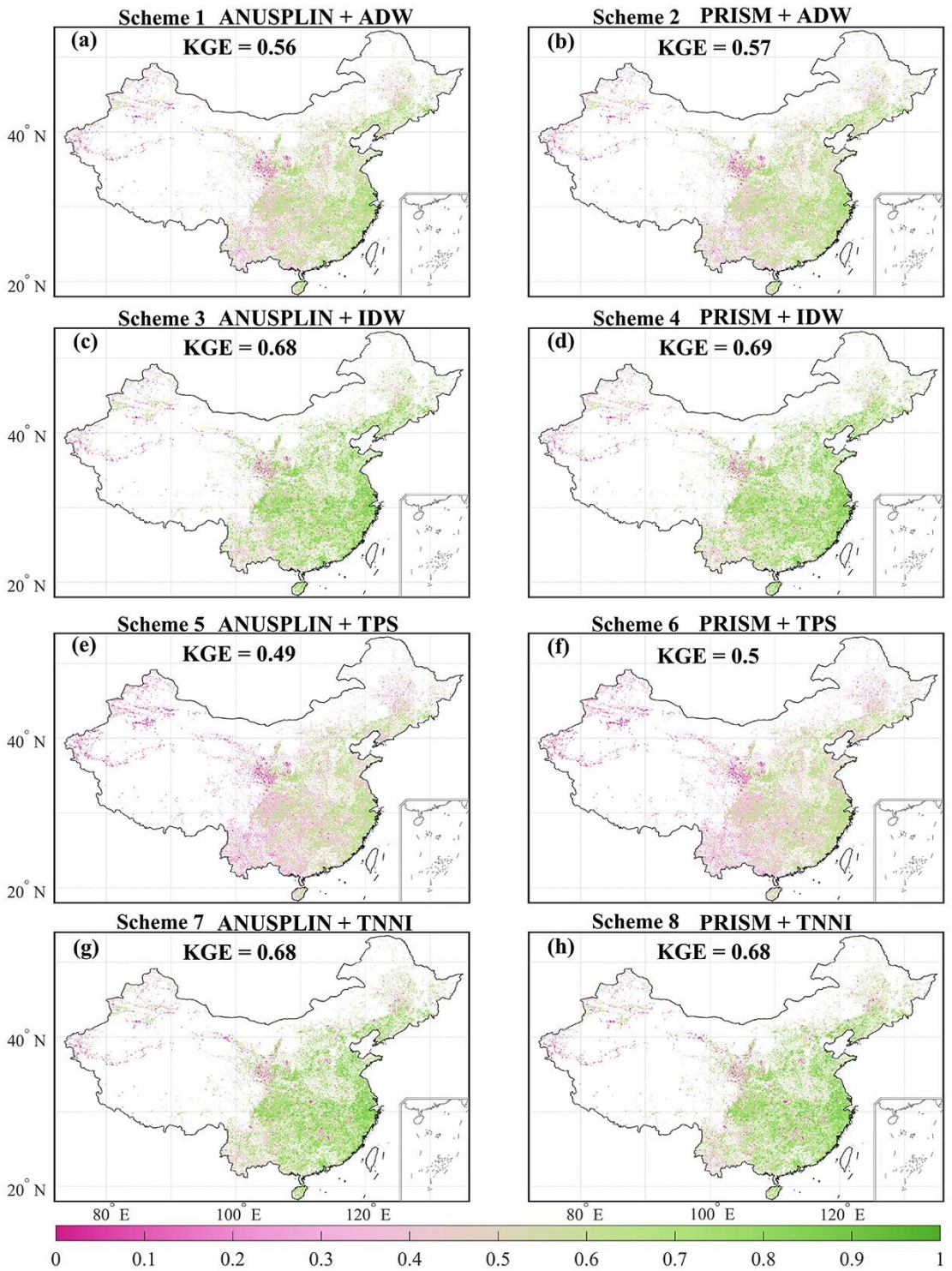

Figure 8. The same as Figure 6, but for Kling-Gupta efficiency (KGE).


## 4.2 Comparison with other gauge-based datasets

We compared the performance of the CHM_PRE dataset with other three datasets—CGDPA (Shen et al., 2010), CN05.1 (Wu and Gao, 2013), and CMA V2.0 (Zhao et al., 2014) (Figures 9–12). To derive a uniform time span among the different datasets,

monthly precipitation series for the period of January 1, 2008, to December 31, 2015, were calculated in the CHM_PRE, CGDPA, CN05.1, and CMA V2.0 datasets. Results showed that the temporal pattern of the monthly precipitation series was generally consistent among different datasets, with a maximum bias of 5 mm/month for the dry (December-January-February) and wet (June-July-August) seasons (Figure 9). The average annual wet-day (>1 mm/day) precipitation amount and frequency between 2008 and 2015 for different datasets shared similar spatial patterns, with a general decrease from southeastern China to northwestern China (Figures 10, 11). The median differences in the multi-year annual wet-day precipitation amount across all grid cells for CHM_PRE − CGDPA, CHM_PRE − CN05.1, and CHM_PRE − CMA V2.0 were −24.79 mm/yr, −7.43 mm/yr, and 13.87 mm/yr, respectively. The multi-year annual wet-day frequency was higher in CGDPA and CMA V2.0 than in the CHM_PRE dataset, with median differences of 6.38 days/yr and 3.25 days/yr across grid cells, respectively. The mean annual wet-day frequency in CN05.1 was 9.63 days/yr less than in the CHM_PRE dataset. The broad features of mean annual maximum 1-day precipitation amount (Rx1day) were comparable among the four datasets from 2008 to 2015 (Figure 12). The 8-year average Rx1day values for China were 48.19 mm/day, 35.29 mm/day, 39.72 mm/day, and 40.19 mm/day for the CGDPA, CN05.1, CMA V2.0, and CHM_PRE datasets, respectively. Generally, the differences in spatial pattern among different datasets were a combined effect of the gauge density involved and whether or not orographic effects and boundary effects were considered in the interpolation algorithm. The spatial patterns of mean precipitation (Figures 10, 11) and extreme precipitation (Figure 12) agreed well among the different datasets in eastern and southern China, where gauge density is relatively high. This indicates the density of input gauges could be a dominant factor affecting interpolation output (Morrissey et al., 1995). Agreement in spatial patterns of mean and extreme precipitation was poorer in northwestern China and the Tibetan Plateau, which was driven by interpolation algorithms. In particular, the heavy precipitation in the southern Tibetan Plateau was well captured by the CGDPA and CHM_PRE datasets, but the CN05.1 and CMA V2.0 datasets failed to capture it. This suggests the importance of orographic effects and

505 boundary effects in interpolation processes, because heavier rainfall appears over mountainous regions than nearby plains.

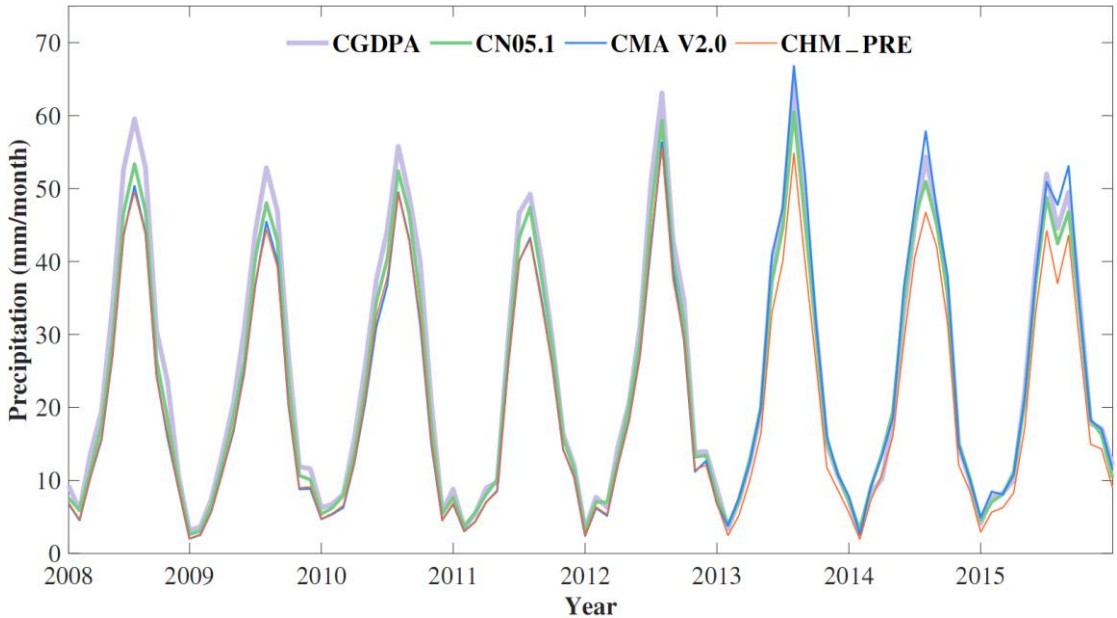

Figure 9. Monthly precipitation series from January 1, 2008, to December 31, 2015, for the CGDPA, CN05.1, CMA V2.0, and CHM_PRE datasets.

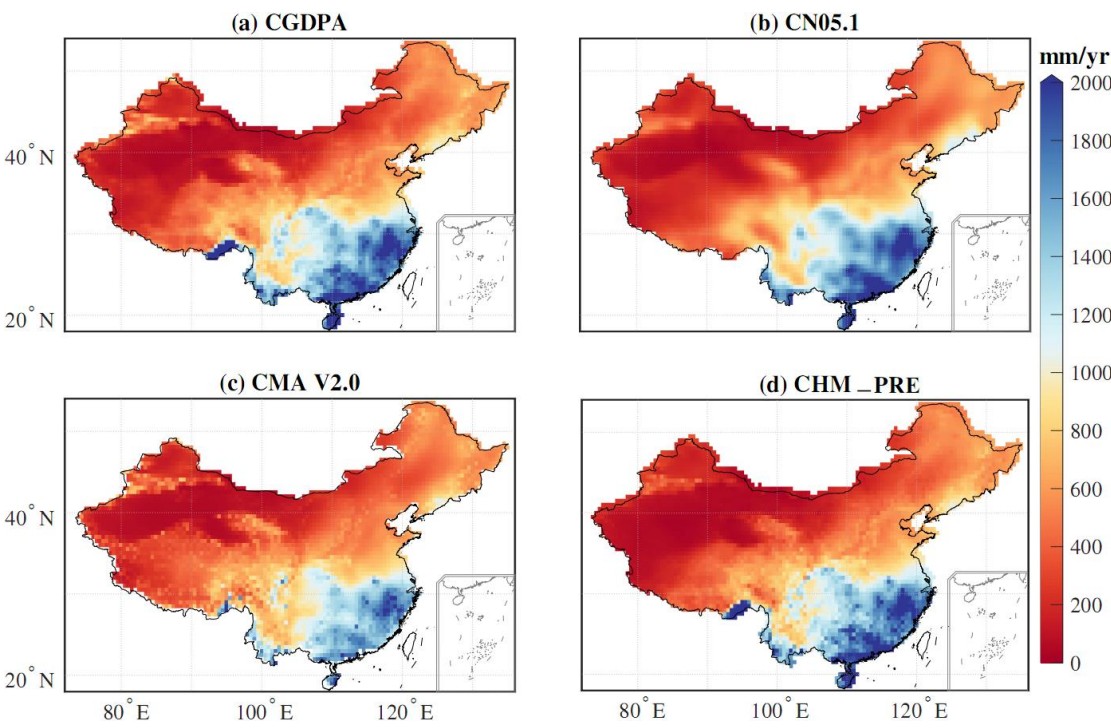

Figure 10. Spatial pattern of average annual wet-day (>1 mm/day) precipitation amount during the period of 2008 to 2015 for the (a) CGDPA, (b) CN05.1, (c) CMA V2.0, and (d) CHM_PRE datasets.

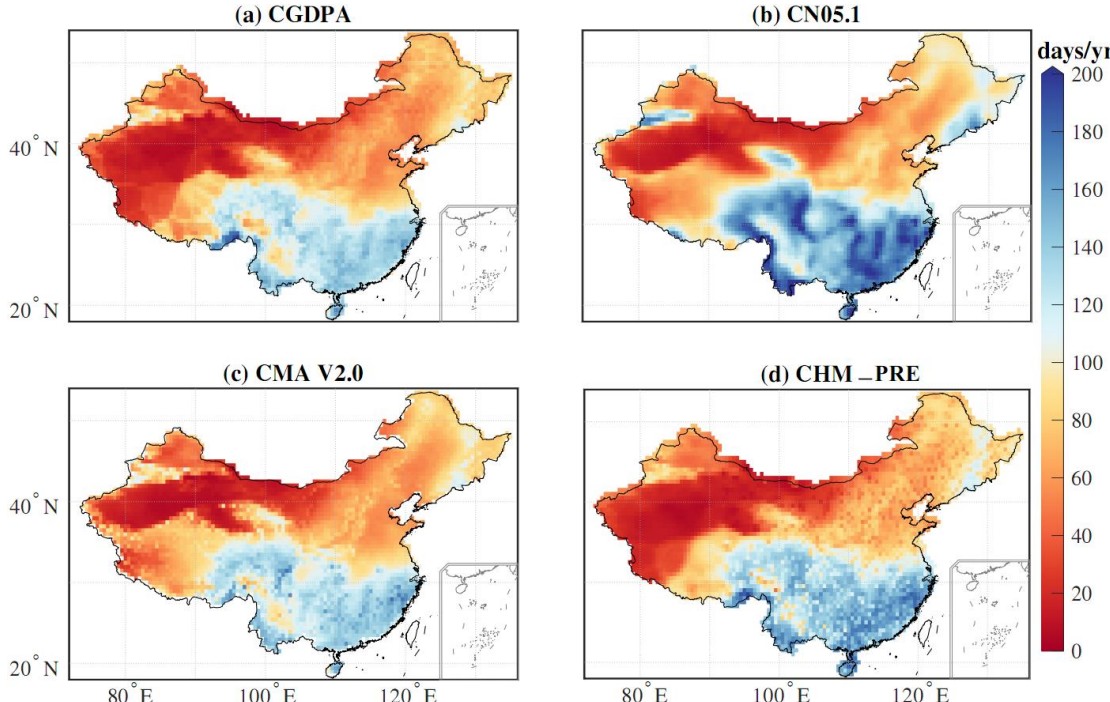

Figure 11. Spatial pattern of mean annual wet-day (>1 mm/day) frequency from 2008 to 2015 for the (a) CGDPA, (b) CN05.1, (c) CMA V2.0, and (d) CHM_PRE datasets.

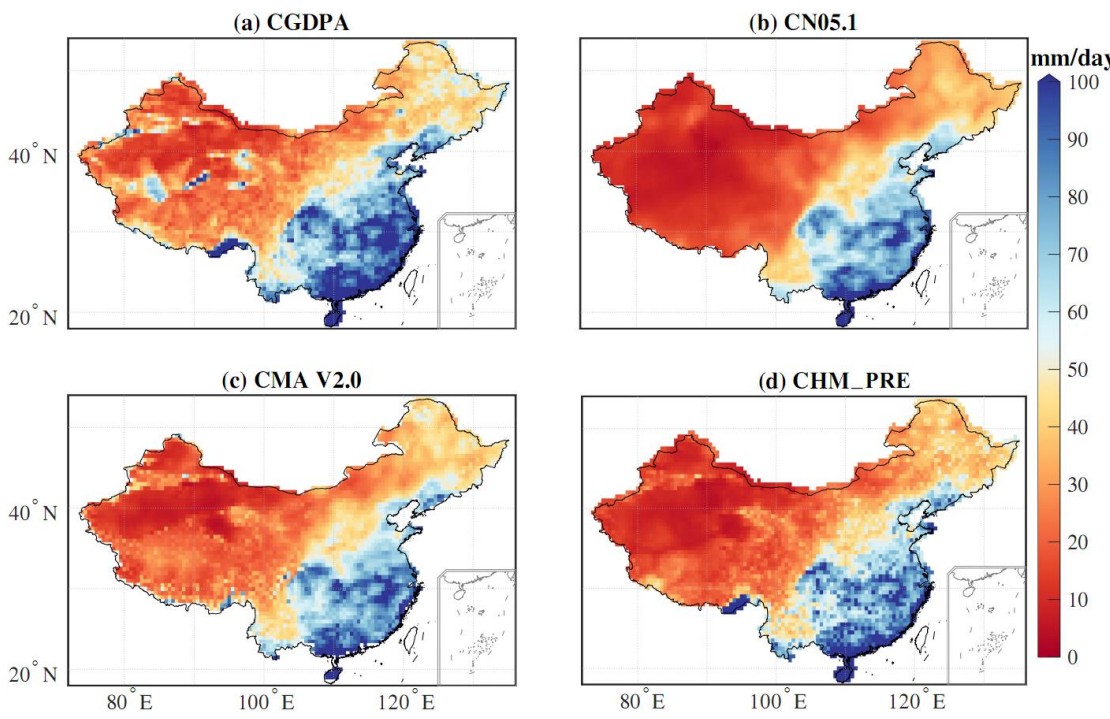

Figure 12. Spatial pattern of mean Rx1day from 2008 to 2015 for the (a) CGDPA, (b) CN05.1, (c) CMA V2.0, and (d) CHM_PRE datasets.

**5 Data availability**

This high-resolution long-term gauge-based daily precipitation dataset covers the period of 1961–2022, and it will continue to be updated annually. It contains data for three spatial resolutions: 0.1° × 0.1°, 0.25° × 0.25° and 0.5° × 0.5°covering the domain of 18°N–54°N, 72°E–136°E. The NetCDF-formatted output files of the CHM_PRE dataset are freely accessible at https://doi.org/10.6084/m9.figshare.21432123.v4 (Han and Miao, 2022).

**6 Conclusions**

Based on a recent 62-yr time series of daily observations from 2,839 gauges across the Chinese mainland and the areas just outside China's boundaries, this study compared eight different interpolation schemes that used an algorithm combining the daily climatology field with a precipitation ratio field. A validation method was used to evaluate the eight interpolation schemes using 45,992 high-density gauge observations from China. The results indicate that the best-performing scheme was scheme 4, which combined a monthly precipitation constraint and correction for topographic characteristics with the daily climatology field and interpolated station observations of precipitation ratio into grid cells using an inverse distance weighting method. The median CC, RMSE, and KGE values for the interpolation scheme that performed the best among the selected metrics (in comparison with the high-density gauge observations used for validation) were 0.78, 8.8 mm/d, and 0.69, respectively. Using the best-performing interpolation scheme, we constructed a new gridded precipitation dataset (CHM_PRE) for the Chinese mainland with a daily temporal resolution and at multiple spatial resolutions (0.1° × 0.1°, 0.25° × 0.25°, and 0.5° × 0.5°) for the period of 1961–2022. The CHM_PRE dataset showed reliable quality compared with other available precipitation products.

**Author Contributions**

JH and CM contributed to designing the research; JH implemented the research and wrote the original draft; CM and JG supervised the research; all co-authors revised the

manuscript and contributed to the writing.

**Competing Interests**

The authors declare that they have no conflict of interest.

**Acknowledgments**

We would like to thank the high-performance computing support from the Center for Geodata and Analysis, Faculty of Geographical Science, Beijing Normal University (https://gda.bnu.edu.cn/). We are also grateful to the National Meteorological Information Center of the China Meteorological Administration (NMIC, http://data.cma.cn) for providing the observed climate data.

**Financial support**

This research was supported by the National Natural Science Foundation of China (No. 42041006), the Second Tibetan Plateau Scientific Expedition and Research Program (STEP) (No.2019QZKK0405), and the State Key Laboratory of Earth Surface 570 Processes and Resource Ecology (2022-ZD-03).

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
