# Peer review of "A new daily gridded precipitation dataset for the Chinese mainland based on gauge observations"

_Earth System Science Data, 2022_

## Referee Comment (RC1)

'A new daily gridded precipitation dataset…' by Han et al.

The authors have produced a grid-based precipitation data product for mainland China. The authors used long-term daily precipitation data from 2419 stations in China since 1961 for interpolation, and additional short-term (2015-2019) gauge data for a much larger number of stations (>40,000) for evaluating 8 different spatial interpolation schemes. The analysis was extensive, and rigorous, and the recommended 'optimal' scheme is justified and well supported with empirical evidence, and final data products at 3 different spatial resolutions will no doubt be most useful for wider applications.

The manuscript is mostly readable, not difficult to follow. English expressions at times are a bit odd, so is the tense.

Overall, the manuscript along with the data product(s) are publishable, with additional effort to improve the clarity and quality of presentation.

Major comments/suggestions:

About spatial resolution: I wonder about the wisdom of making data products available at all 3 spatial resolutions, namely, 0.1, 0.25, and 0.5 degree. Users of these products surely would be able to resample the dataset at finer resolution to coarse ones. I also wonder why data product at 0.05-degree resolution was not attempted since both gridded daily and monthly climatologies at 0.05 were used for this manuscript. 0.05-degree is commonly for areas of comparable size such as Australia (Jeffrey et al. 2001), and datasets at much finer spatial resolution (0.01-degree) are available for densely gauged areas such as Japan (Hatono et al. 2022).

Can the authors make it absolutely clear whether the gridded data refer to point precipitation at the centre of each grid cell, or to areal average precipitation over the grid cell. This has considerable implications for how these datasets are used and/or resampled, especially when the centres of grid cells of differing resolutions are co-located.

There were fewer stations, up to 15%, over the two two decades (1961-1980) for interpolated precipitation products (Fig. 1). This warrants discussion towards the end of manuscript.

Clarify the notion of 'daily climatology' as this term is not widely used and understood. Based on my understanding of the manuscript, there are three daily climatologies:

Raw daily climatology – simply mean daily precipitation amount (mm/d) for the period (1971-2000)

Smoothed daily climatology – high frequency fluctuations are removed with only the first few harmonics retained. (the authored all this the raw daily climatology, but why?)

Adjusted daily climatology – the smoothed daily precipitation was adjusted proportionally so that the monthly precipitation was preserved.

I would recommend use of the mean daily precipitation amount instead of 'daily climatology'. It is much easier to understand.

Discarding high-frequency 'noise' in the mean daily precipitation (line 202-204) would lead to a reduction in total variation in the daily climatology. How much variation preserved relative to the total variation in the mean daily precipitation? Please indicate a range.

There are issues with most of the equations used in the manuscripts see comments below

Check the tense used throughout the manuscript. Use the past tense to describe what you did, and present or present perfect to describe what others have said or done.

Minor comments/edits (the original in black, revised in blue)

Line 43: warmer at Earth's surface -> warmer at the Earth's surface

Line 44: between the atmosphere and surface -> between the atmosphere and land surface

Line 52: dataset is essential to current hydrometeorology research -> dataset is essential for hydrometeorological research

Line 55: The measurement of precipitation relies mainly on direct measurement using rain gauges disdrometers, and radar and on indirect estimation using satellite systems. -> Collection of precipitation data relies mainly on measurements using ground-based rain gauges, and estimates using sensing technologies such as weather radar and satellite.

Line 60: However, gauge observations reflect only point precipitation, and -> However, precipitation data measured with gauges are point observations only, and

Line 67 Spatial interpolation methods are usually applied to convert irregular point observations to regional measurements (Ahrens, 2006), thus generating evenly gridded precipitation products that are widely used in hydrology and 70 meteorology studies (Schamm et al., 2014; Golian et al., 2019).

Spatial interpolation methods are usually applied to irregular point observations to produce evenly distributed precipitation grid (Ahrens, 2006) for application in hydrological and meteorological studies (Schamm et al., 2014; Golian et al., 2019).

Line 78: To reach a higher temporal resolution, a daily gridded precipitation dataset has been built across China using the same raw precipitation data with a time span from 80 1961 to 2019 (Qin et al., 2022).

To achieve a higher temporal resolution, a daily gridded precipitation dataset has been produced for China using the same raw precipitation data for the same period from 1961 to 2019 (Qin et al., 2022).

(NB, 'across China' is used extensively in the manuscript. The word 'across' suggests 'from one side to another', and the word may not be appropriate in all cases. Please review its usage in the manuscript.)

Line 99 boundaries suffer worse positioning accuracy relative - > boundaries suffer positioning inaccuracy

Line 111: Eight interpolation schemes are proposed and evaluated by cross validation -> Eight interpolation schemes were considered and evaluated with cross validation

Line 115 is provided publicly for applications -> is available in the public domain

Line 121 are collected -> were collected.

Line 124 available to use for -> available for

Line 126: The coverage of stations is relatively sparse over northwestern China -> Stations are sparsely distributed in northwestern China

Line 140 stations with a range of 2015–2019 across China -> stations for the period 2015–2019 in China

Line 142
After removing stations with a missing
rate of over 20% for the period of 2015–2019, 45,992 good-quality stations are left (Figure 1b).

Once stations with more than 20% missing data were removed, there were 45,992 good-quality stations available for cross validation (Figure 1b).

Line 152: is -> was

Line 165:

for the following climatology adjustment.

for adjustment based on climatology

Line 217: the monthly total of the raw gridded daily climatology field for

(unclear, is this the sum of the smoothed mean daily precipitation for the month?)

Line 224

Equation (1)

(Something is not quite correct here. There is no index j on the right hand side. Is SF value the same for every day of the month? In other words, are there only 12 distinct SF values for each year?)

Line 255:

we used in -> used for

Equation (4)

Should be expressed as e^(-nx_i/GDD)

Equation (5)

What is the definition of surrounding stations?

Equation (6)

Normally we do not use 'X' in equations to indicate multiplication.

For IDW, again what are the surrounding stations?

Line 247 – repetition from data section above.

References

Hatono et al. 2022. Scientific Data. https://doi.org/10.1038/s41597-022-01548-3

Jeffrey, S.J., Carter, J.O., Moodie, K.B., Beswick, A.R., 2001. Using spatial interpolation to construct a comprehensive archive of Australian climate data. Environ. Modelling and Software 16 (4), 309–330.

---

## Author Comment (AC1)

The authors have produced a grid-based precipitation data product for mainland China. The authors used long-term daily precipitation data from 2419 stations in China since 1961 for interpolation, and additional short-term (2015-2019) gauge data for a much larger number of stations (>40,000) for evaluating 8 different spatial interpolation schemes. The analysis was extensive, and rigorous, and the recommended 'optimal' scheme is justified and well supported with empirical evidence, and final data products at 3 different spatial resolutions will no doubt be most useful for wider applications.

The manuscript is mostly readable, not difficult to follow. English expressions at times are a bit odd, so is the tense.

Overall, the manuscript along with the data product(s) are publishable, with additional effort to improve the clarity and quality of presentation.

**Response:** We would like to thank you for your constructive comments on our manuscript. Your insightful review has enhanced our paper considerably. We have updated the dataset to 2022 (https://doi.org/10.6084/m9.figshare.21432123.v3). Below is a point-by-point response to your comments.

Major comments/suggestions:

About spatial resolution: I wonder about the wisdom of making data products available at all 3 spatial resolutions, namely, 0.1, 0.25, and 0.5 degree. Users of these products surely would be able to resample the dataset at finer resolution to coarse ones. I also wonder why data product at 0.05-degree resolution was not attempted since both gridded daily and monthly climatologies at 0.05 were used for this manuscript. 0.05-degree is commonly for areas of comparable size such as Australia (Jeffrey et al. 2001), and datasets at much finer spatial resolution (0.01-degree) are available for densely gauged areas such as Japan (Hatono et al. 2022).

**Response:** Thanks for your comments. In consideration of users' research demands for various spatial resolutions, CHM_PR provides daily precipitation series with multiple spatial resolutions

so that users can easily find the best matches. Meanwhile, as our strategy for creating products, the daily precipitation is first produced on a 0.05-degree latitude-longitude grid. Computing the analyzed values (here, daily precipitation) at a finer resolution enables improved correction of the orographic effects, which exhibit rapid changes with elevation (Daly et al. 1994; Xie et al. 2007). Also, this makes it convenient to generate analyzed values at various resolutions for different applications.

The main reason for not including a spatial resolution of 0.05-degree in the dataset is due to the prevailing demand for daily precipitation data in China and the limitations of gauge density. In China's hydrometeorology research, daily precipitation with spatial resolutions of 0.1-degree, 0.25-degree, and 0.5-degree are the most widely used. Additionally, when considering gauge density, only 0.29% of all 0.05-degree grid cells contain at least one station out of the 2,839 available gauges. This may not provide sufficient support for estimating daily precipitation at a high resolution of 0.05 degrees.

Can the authors make it absolutely clear whether the gridded data refer to point precipitation at the centre of each grid cell, or to areal average precipitation over the grid cell. This has considerable implications for how these datasets are used and/or resampled, especially when the centres of grid cells of differing resolutions are co-located.

**Response:** Thank you for your insightful suggestion. The gridded data use the areal average precipitation over the grid cell. We have added this clarification to section 4.1.

There were fewer stations, up to 15%, over the two decades (1961-1980) for interpolated precipitation products (Fig. 1). This warrants discussion towards the end of manuscript.

**Response:** Thanks for pointing this out. Due to a high rate of missing daily observations from 1961 to 1980, up to 15% of stations could not reach the threshold for quality control (i.e. a rate of missing daily precipitation not more than 5%) and were removed. With the vigorous development of hydrometeorological observation in China since the 1980s, precipitation data quality has been getting better and better, and the missing rate has been getting lower and lower, which caused a jump in the number of stations that met this quality control requirement beginning in 1981 (Shen et al., 2014). If we had kept the number of gauges used for interpolation steady over the full 62

year-span, about 300 gauges available for the period 1981–2022 would have been excluded, which would have been a great loss of real observed precipitation information over the last 40 years. Therefore, the strategy adopted in this study was that all observational data that met the quality control conditions were used for data interpolation, which led to some differences in the number of sites used every year. These slight differences could be partly compensated for by using correlation decay distance (CDD1 and CDD2), which confirmed there were at least three stations involved in the interpolation for each grid cell so that there would not be a sharp change in the number of stations used for interpolation for each grid cell.

We have added discussion about this in Section 4.1 of the manuscript as follows:
"Due to a high rate of missing daily observations from 1961 to 1980, up to 15% of stations could not reach the threshold for quality control (i.e., a rate of missing daily precipitation not more than 5%) and were removed. With the vigorous development of hydrometeorological observation in China since the 1980s, precipitation data quality has been improving, which caused a jump in the number of stations that met quality control requirements beginning in 1981 (Shen et al., 2014). If we had kept the number of gauges used for interpolation steady over the full 62 year-span, about 300 gauges available for the period 1981–2022 would have been excluded, which would have been a great loss of real observed precipitation information. Therefore, the strategy adopted in this study was that all observational data that met the quality control conditions were used for data interpolation, which led to some differences in the number of sites used every year. These slight differences could be partly compensated for by using correlation decay distance (CDD1 and CDD2), which confirmed there were at least three stations involved in the interpolation for each grid cell so that there would not be a sharp change in the number of stations used for interpolation for each grid cell."

Clarify the notion of 'daily climatology' as this term is not widely used and understood. Based on my understanding of the manuscript, there are three daily climatologies:
Raw daily climatology – simply mean daily precipitation amount (mm/d) for the period (1971-2000);
Smoothed daily climatology – high frequency fluctuations are removed with only the first few harmonics retained. (the authored all this the raw daily climatology, but why?);

Adjusted daily climatology – the smoothed daily precipitation was adjusted proportionally so that the monthly precipitation was preserved.

I would recommend use of the mean daily precipitation amount instead of 'daily climatology'. It is much easier to understand.

**Response:** Thanks for your constructive suggestions. The definition of "daily climatology" is the mean value for each day over a specified time range (https://iridl.ldeo.columbia.edu/dochelp/StatTutorial/Climatologies/index.html). In this study, it does refer to the mean daily precipitation amount. Because this term is frequently employed in atmospheric sciences, we would like to keep the term "daily climatology" in the manuscript. We have revised the phrases related to the term "daily climatology" according to your suggestions and to better clarify the use of gauges versus grid cells. The revised phrases we used in this study are divided into three categories:

1) Gauge-based climatology of daily precipitation, which is defined as the Fourier-truncated 30-year mean daily precipitation series produced from gauge observations for the period of 1971–2000 for each of the 365 calendar days. We revised the phrase "raw daily climatology" in the original version into "gauge-based climatology of daily precipitation" to better convey that the process is based on gauge observations instead of grid cells. To produce the gauge-based climatology of daily precipitation, the simple mean daily precipitation amounts (mm/d) for the period (1971–2000) are first calculated for each gauge, and then high-frequency fluctuations are removed, with only the first few harmonics retained by Fourier truncation. The revised expressions for this phrase in the manuscript are listed as follows:

"First, the gauge-based climatology of daily precipitation was calculated using gauge observations. The definition of gauge-based climatology of daily precipitation is the Fourier-truncated 30-year mean daily precipitation series produced from gauge observations for the period of 1971–2000 for each of the 365 calendar days (Figure 4). We used Fourier truncation to remove the high-frequency noise of the 30-year mean daily precipitation series for each station and retained the accumulation of the first six harmonic components as the gauge-based climatology of daily precipitation (Xie et al., 2007). After Fourier truncation, approximately 75% of all stations preserve a variation of 40% to 75% in the truncated mean daily precipitation series relative to the total variation in the mean daily precipitation."

2) The unadjusted 0.05° × 0.05° gridded daily climatology field, which is interpolated from the gauge-based climatology of daily precipitation with SRTM-DEM as a covariate using ANUSPLIN software. We have changed the phrase "raw 0.05° × 0.05° gridded daily climatology field" in the previous manuscript into the phrase "unadjusted 0.05° × 0.05° gridded daily climatology field" to emphasize this is the original 0.05° × 0.05° gridded daily climatology field without any adjustment processes applied, such as monthly precipitation constraint or topographic characteristic correction. We have altered the corresponding explanation for this phrase in the manuscript as follows:

"The unadjusted 0.05° × 0.05° gridded daily climatology field ($Cd_0$) was then interpolated from the gauge-based climatology of daily precipitation with SRTM-DEM as a covariate using the ANUSPLIN software (Hutchinson and Xu, 2004)."

3) The adjusted gridded daily climatology field, which was adjusted by using the monthly climatology field to consider a monthly precipitation constraint and topographic characteristic correction. We have added this expression ("the adjusted gridded daily climatology field") into the manuscript as follows:

"To minimize systematic bias from the unadjusted 0.05° × 0.05° gridded daily climatology field on the monthly climatology field ($Cm$), the monthly accumulation of the unadjusted 0.05° × 0.05° gridded daily climatology field was then constrained by the monthly climatology field. This produced an adjusted gridded daily climatology field that uses a monthly precipitation constraint and topographic characteristic correction."

Discarding high-frequency 'noise' in the mean daily precipitation (line 202-204) would lead to a reduction in total variation in the daily climatology. How much variation preserved relative to the total variation in the mean daily precipitation? Please indicate a range.

**Response:** Many thanks for your advice. After Fourier truncation, approximately 75% of all stations preserve a variation of 40% to 75% in the truncated mean daily precipitation series relative to the total variation in the mean daily precipitation. We have added this expression to Section 3.2.

Check the tense used throughout the manuscript. Use the past tense to describe what you did, and present or present perfect to describe what others have said or done.

**Response:** Thank you for your suggestion. We have carefully checked and revised the tense.

Minor comments/edits (the original in black, revised in blue)

Line 43: warmer at Earth's surface -> warmer at the Earth's surface
**Response:** Done.

Line 44: between the atmosphere and surface -> between the atmosphere and land surface
**Response:** Done.

Line 52: dataset is essential to current hydrometeorology research -> dataset is essential for hydrometeorological research
**Response:** Done.

Line 55: The measurement of precipitation relies mainly on direct measurement using rain gauges disdrometers, and radar and on indirect estimation using satellite systems. -> Collection of precipitation data relies mainly on measurements using ground-based rain gauges, and estimates using sensing technologies such as weather radar and satellite.
**Response:** Done.

Line 60: However, gauge observations reflect only point precipitation, and -> However, precipitation data measured with gauges are point observations only, and
**Response:** Done.

Line 67: Spatial interpolation methods are usually applied to convert irregular point observations to regional measurements (Ahrens, 2006), thus generating evenly gridded precipitation products that are widely used in hydrology and 70 meteorology studies (Schamm et al., 2014; Golian et al., 2019).
Spatial interpolation methods are usually applied to irregular point observations to produce evenly distributed precipitation grid (Ahrens, 2006) for application in hydrological and meteorological studies (Schamm et al., 2014; Golian et al., 2019).

**Response:** Done.

Line 78: To reach a higher temporal resolution, a daily gridded precipitation dataset has been built across China using the same raw precipitation data with a time span from 1961 to 2019 (Qin et al., 2022).

To achieve a higher temporal resolution, a daily gridded precipitation dataset has been produced for China using the same raw precipitation data for the same period from 1961 to 2019 (Qin et al., 2022).

(NB, 'across China' is used extensively in the manuscript. The word 'across' suggests 'from one side to another', and the word may not be appropriate in all cases. Please review its usage in the manuscript.)

**Response:** Done.

Line 99 boundaries suffer worse positioning accuracy relative - > boundaries suffer positioning inaccuracy

**Response:** Done.

Line 111: Eight interpolation schemes are proposed and evaluated by cross validation -> Eight interpolation schemes were considered and evaluated with cross validation

**Response:** Since the phrase "cross validation" is not appropriate here, we have revised the sentence as follows:

"Eight interpolation schemes were considered and validated using 45,992 gauge observations for the period of 2015–2019 over China."

Line 115 is provided publicly for applications -> is available in the public domain

**Response:** Done.

Line 121 are collected -> were collected.

**Response:** Done.

Line 124 available to use for -> available for

**Response:** Done.

Line 126: The coverage of stations is relatively sparse over northwestern China -> Stations are sparsely distributed in northwestern China

**Response:** Done.

Line 140 stations with a range of 2015–2019 across China -> stations for the period 2015–2019 in China

**Response:** Done.

Line 142

After removing stations with a missing rate of over 20% for the period of 2015–2019, 45,992 good-quality stations are left (Figure 1b).

Once stations with more than 20% missing data were removed, there were 45,992 good-quality stations available for cross validation (Figure 1b).

**Response:** Done. Also, since the term "cross validation" is not appropriate here, we have revised it to "validation".

Line 152: is -> was

**Response:** Done.

Line 165:

for the following climatology adjustment. -> for adjustment based on climatology

**Response:** Done.

Line 217: the monthly total of the raw gridded daily climatology field for

(unclear, is this the sum of the smoothed mean daily precipitation for the month?)

**Response:** The monthly total of the unadjusted $0.05° \times 0.05°$ gridded daily climatology field is

derived by taking the sum of the unadjusted 0.05° × 0.05° gridded daily climatology field for the month. This process uses the gridded daily climatology field, which is interpolated from the gauge-based climatology of daily precipitation that has experienced Fourier truncation.

We have revised the sentence as follows: "Calculate $Cd_{0\_(m,j)}$ ($m$ = 1, 2, 3, …, 12; $j$ = 1, 2, 3, …, 365; $m$ is the corresponding month for day $j$), which is the monthly total of the unadjusted 0.05° × 0.05° gridded daily climatology field, derived by taking the sum of the unadjusted 0.05° × 0.05° gridded daily climatology field for the month."

Line 224, Equation (1)
(Something is not quite correct here. There is no index j on the right hand side. Is SF value the same for every day of the month? In other words, are there only 12 distinct SF values for each year?)

**Response:** Thanks for your keen observation. We have checked and revised equations (1) and (2) like this:

"3) Compute the scaling factor $SF_{(m,j)}$ for the individual calendar day of the unadjusted 0.05° × 0.05° daily climatology field to the gridded monthly climatology field:

$$SF_{(m,j)} = \frac{C_{(m,j)}}{w_{(m-1,j)}\, Cd_{0\_(m-1,j)} + w_{(m,j)}\, Cd_{0\_(m,j)} + w_{(m+1,j)}\, Cd_{0\_(m+1,j)}} \tag{1}$$

$$(m = 1, 2, 3, …, 11, 12;\ j = 1, 2, 3, …, 365;$$
$$m \text{ is the corresponding month for day } j)$$

where $C_{(m,j)}$ is the gridded monthly climatology field for the corresponding month $m$ of day $j$; $Cd_{0\_(m-1,j)}$, $Cd_{0\_(m,j)}$ and $Cd_{0\_(m+1,j)}$ are the monthly total of months $m-1$, $m$, and $m+1$, respectively, which are calculated from the unadjusted 0.05° × 0.05° gridded daily climatology field; $w_{(m-1,j)}$, $w_{(m,j)}$, and $w_{(m+1,j)}$ are the corresponding weights for months $m-1$, $m$, and $m+1$, respectively, which are inversely proportional to the interval between the calendar day $j$ and the center of the month (Xie et al., 2007). Note that the weight $w_{(m-1,j)}$ is zero when $m = 1$, and so is the weight $w_{(m+1,j)}$ when $m = 12$.

4) The adjusted gridded daily climatology field ($Cd_{(m,j)}$) is defined as

$$Cd_{(m,j)} = Cd_{0\_(m,j)}\, SF_{(m,j)} \tag{2}$$

$$(m = 1, 2, 3, \dots, 11, 12; \ j = 1, 2, 3, \dots, 365;$$

$m$ is the corresponding month for day $j$)"

Line 255: we used in -> used for

**Response:** Done.

Equation (4), Should be expressed as e^(-nx_i/GDD)

**Response:** Thanks for the suggestion. We have added the subscript $(x_i)$ to represent the distance between station $i$ and the center of target grid cell L. To make it easier for readers to understand, we keep the power $n$ outside of the bracket. The expression for equation (4) has been revised as follows:

$$\text{``}D_i = \left(e^{-x_i/CDD}\right)^n \tag{4}$$

where $x_i$ is the distance between station $i$ and the center of target grid cell L; $n$ is a constant and usually set to 4, in accordance with previous studies (Harris et al., 2020; Dunn et al., 2020; Efthymiadis et al., 2006)."

Equation (5), What is the definition of surrounding stations?

**Response:** "Surrounding stations" are all of the stations except for station $i$. We have added this definition in the manuscript: 'Here, "surrounding stations" refers to all other stations except for station $i$.'

Equation (6), Normally we do not use 'X' in equations to indicate multiplication.

**Response:** Many thanks for the advice. We have revised the equation (6) into the following format: "$W_i = D_i A_i$"

For IDW, again what are the surrounding stations?

**Response:** For IDW, the surrounding stations are defined as the stations that fall in the search radius of the target grid cell.

We have added the definition in the manuscript as follows: "We employed $S_0$ representing the target grid cell, $i$ representing the surrounding stations that fall in the search radius of the target grid cell, $y(s_i)$ denoting the station observations, and $d_{0i}$ denoting the distance between $S_0$ and $i$."

Line 247 – repetition from data section above.

**Response:** Done.

References

Hatono et al. 2022. Scientific Data. https://doi.org/10.1038/s41597-022-01548-3

Jeffrey, S.J., Carter, J.O., Moodie, K.B., Beswick, A.R., 2001. Using spatial interpolation to construct a comprehensive archive of Australian climate data. Environ. Modelling and Software 16 (4), 309–330.

References:

Daly, C., Neilson, R. P., and Phillips, D. L.: A statistical-topographic model for mapping climatological precipitation over mountainous terrain, Journal of Applied Meteorology, 33, 140–158, https://doi.org/10.1175/1520-0450(1994)033<0140:ASTMFM>2.0.CO;2, 1994.

Shen, Y., Zhao, P., Pan, Y., and Yu, J.: A high spatiotemporal gauge-satellite merged precipitation analysis over China, Journal of Geophysical Research: Atmospheres, 119, 3063–3075, https://doi.org/10.1002/2013JD020686, 2014.

Xie, P., Chen, M., Yang, S., Yatagai, A., Hayasaka, T., Fukushima, Y., and Liu, C.: A gauge-based analysis of daily precipitation over east Asia, Journal of Hydrometeorology, 8, 607–626, https://doi.org/10.1175/JHM583.1, 2007.

---

## Author Comment (AC2)

**A DETAILED LIST OF RESPONSES**
**TO REVIEWER #2**

The objective of the manuscript is to present a new gridded precipitation dataset across mainland China using the best interpolation scheme among the 8 tested. Reliable precipitation data are important to ensure water safety and guarantee water availability and quality. Hence, efforts in creating reliable datasets are quite valuable.

**Response:** We would like to thank you for your constructive comments on our manuscript. Your insightful review has enhanced our paper considerably. We have updated the dataset to 2022 (https://doi.org/10.6084/m9.figshare.21432123.v3). Below is a point-by-point response to your comments.

However, the manuscript in its current form lacks a critical discussion on the limitation of the gridded data available and on the selected interpolation scheme. The Authors provide a list of gridded precipitation datasets available (Lines 71-104). However, a critical review of such datasets is missing. They mentioned the sensitivity to interpolation algorithms, but it is too vague. The Authors do not discuss why the scheme considered the optimal (even though it is not an optimal scheme but rather the best, based on some metrics, among the few schemes tested) leads to better goodness of fit metrics. Is such a result expected? Why is such a combination better than the others? It is simply chance? Can this scheme be transferred to other regions?

**Response:** Many thanks for the insightful comments. We have added Table 1 for a summary of the current daily gridded precipitation datasets over China:

Table 1 Gauge-based gridded precipitation datasets for China

| Name | Spatial resolution | Domain | Temporal resolution | Time period | Reference | Number of stations | Interpolation method |
|---|---|---|---|---|---|---|---|
| 1 km monthly temperature and precipitation dataset for China from 1901 to 2017 | 1 km | China | Monthly | 1901 to the present | Peng et al., 2019 | ~700 | Bilinear interpolation |
| HRLT | 1 km | China | Daily | 1961–2019 | Qin et al., 2022 | ~700 | Machine learning, the generalized additive model, and thin plate spline |
| CMFD | 0.1° × 0.1° | China | Three hours | 1979 to the present | He et al.,2020 | ~700 | Thin plate spline |
| EA05 | 0.5° × 0.5° | East Asia | Daily | 1978–2003 | Xie et al., 2007 | ~1,700 | Optimal interpolation |
| CN05.1 | 0.25° × 0.25° | China | Daily | 1961 to the present | Wu and Gao, 2013 | ~2,400 | Angular distance weight |
| CMA V2.0 | 0.5° × 0.5° | China | Daily | 1961–2019 | Zhao et al., 2014 | ~2,400 | Thin plate spline |
| CGDPA | 0.25° × 0.25°, 0.5° × 0.5° | China | Daily | 2008–2015 | Shen et al., 2010 | ~2,400 | Optimal interpolation |

And instead of the phrase "optimal scheme" used in the original text, we have revised to use "the interpolation scheme with better performance among selected metrics".

In addition, we also have discussed the main reasons for the differences among various gauge-based precipitation datasets in section 4.2. The reasons include the density of gauges involved and whether or not the interpolation scheme fully considers the impact of topography and boundary effects. Considering the spatiotemporal consistency of the daily gauge observations, the differences in interpolation performance among the eight interpolation schemes are mainly driven by whether the interpolation scheme fully considers the impact of topography and boundary effects.

The overall interpolation strategy was to combine the daily climatology field ($Cd$) with the field of the ratio between daily precipitation and daily climatology ($P/Cd$). With respect to $Cd$, the PRISM-type daily climatology field incorporates topographic features, proximity to coastlines, and several measures of terrain complexity, which goes beyond the climate–elevation relationships that the ANUSPLIN-type daily climatology field considers. As for $P/Cd$, we selected the four alternative interpolation methods (angular-distance weighting (ADW), inverse distance weighting (IDW), thin plate spline (TPS), and triangulation-based nearest neighbor interpolation (TNNI)) to consider the balance between local data fidelity and global fitting smoothness in addition to the popularity, authority, and simplicity of the interpolation methods. Specifically, the ADW and IDW methods were chosen due to their high local data fidelity. Both are local interpolation methods (Liszka, 1984). Unlike the IDW method, the ADW method assigns a tiny weight to far-distant gauge observations to promote global fitting smoothness. This impacts the local accuracy of interpolation. The TPS and TNNI methods, on the other hand, are chosen for their high global fitting smoothness. Both TPS and TNNI are global interpolation methods (Liszka, 1984). The TPS method is based on a mathematical model for surface estimation that fits a minimum-curvature surface through all input points, while TNNI constructs a Delaunay triangulation of three station locations. So TNNI tends to assign more weights to maintain local data fidelity but has weaker fitting smoothness. To sum up, the combination of PRISM-type $Cd$ and IDW-type $P/Cd$ yields the best performance among the selected schemes. This is not simply due to chance. This best-performing interpolation scheme could be applied in other regions, but further validation would be needed to confirm whether it is the best-performing interpolation scheme there. We have added

relevant discussion on the best-performing scheme in the Section 4.1 as follows:

"Scheme 4 had better performance than the other schemes because it considers the impact of topography more deeply and holds an appropriate balance between local data fidelity and global fitting smoothness. The overall interpolation strategy was to combine the daily climatology field ($Cd$) with the field of the ratio between daily precipitation and daily climatology ($P/Cd$). With respect to $Cd$, the PRISM-type daily climatology field incorporates topographic features, proximity to coastlines, and several measures of terrain complexity, which goes beyond the climate–elevation relationships that the ANUSPLIN-type daily climatology field considers. As for $P/Cd$, we selected the four alternative interpolation methods (angular-distance weighting (ADW), inverse distance weighting (IDW), thin plate spline (TPS), and triangulation-based nearest neighbor interpolation (TNNI)) to consider the balance between local data fidelity and global fitting smoothness in addition to the popularity, authority, and simplicity of the interpolation methods. Specifically, the ADW and IDW methods were chosen due to their high local data fidelity. Both are local interpolation methods (Liszka, 1984). Unlike the IDW method, the ADW method assigns a tiny weight to far-distant gauge observations to promote global fitting smoothness. This impacts the local accuracy of interpolation. The TPS and TNNI methods, on the other hand, were chosen for their high global fitting smoothness. Both TPS and TNNI are global interpolation methods (Liszka, 1984). The TPS method is based on a mathematical model for surface estimation that fits a minimum-curvature surface through all input points, while TNNI constructs a Delaunay triangulation of three stations locations. So TNNI tends to assign more weights to maintain local data fidelity but has weaker fitting smoothness. To sum up, the combination of PRISM-type $Cd$ and IDW-type $P/Cd$ yielded the best performance among the selected schemes. This was not simply due to chance. This best-performing interpolation scheme could be applied in other regions, but further validation would be needed to confirm whether it is the best-performing interpolation scheme there."

Point-by-point comments:

Abstract: I suggest the Authors revise the abstract. The primary objective of the paper (the new gridded data) and the temporal coverage (from when to when) should be better highlighted. Moreover, it should be clearer why the interpolation method selected is the best among the ones

tested and how it addresses the limitations of currently available products. RMSE and other metrics as presented are not enough to judge the goodness of the method. How does this perform compared to the others? Why does it perform better?

**Response:** Many thanks for your insightful suggestions. We have revised the abstract as follows: "High-quality, freely accessible, long-term precipitation estimates with fine spatiotemporal resolution play essential roles in hydrologic, climatic, and numerical modeling applications. However, the existing daily gridded precipitation datasets over China either are constructed with insufficient gauge observations or neglect topographic effects and boundary effects on interpolation. Using daily observations from 2,839 gauges located across China and nearby regions from 1961 to the present, this study compared eight different interpolation schemes that adjusted the climatology based on a monthly precipitation constraint and topographic characteristic correction, using an algorithm that combined the daily climatology field with a precipitation ratio field. Results from these eight interpolation schemes were validated using 45,992 high-density daily gauge observations from 2015 to 2019 from China. Of these eight schemes, the one with the best performance merges the Parameter-elevation Regression on Independent Slopes Model (PRISM) in the daily climatology field and interpolates station observations into the ratio field using an inverse distance weighting method. This scheme had median values of 0.78 for the correlation coefficient, 8.8 mm/d for the root-mean-square deviation, and 0.69 for the Kling-Gupta efficiency for comparisons between the 45,992 high-density gauge observations and the best interpolation scheme for the 0.1° latitude × longitude grid cells from 2015 to 2019. This scheme had the best overall performance, as it fully considers topographic effects in the daily climatology field and it balances local data fidelity and global fitting smoothness in the interpolation of the precipitation ratio field. Therefore, this scheme was used to construct a new long-term, gauge-based gridded precipitation dataset for the Chinese mainland (called CHM_PR, as a member of the China Hydro-Meteorology dataset) with spatial resolutions of 0.5°, 0.25°, and 0.1° from 1961 to the present. This precipitation dataset is expected to facilitate the advancement of drought monitoring, flood forecasting, and hydrological modeling. Free access to the dataset can be found at https://doi.org/10.6084/m9.figshare.21432123.v3 (Han and Miao, 2022)."

Lines 80-81: what does the following sentence mean? "Through a fusion of remote sensing products and reanalysis datasets into in situ station data". Remote sensing products and reanalysis

data generated gauged precipitation dataset? Or gauged data were combined with remote sensing products and reanalysis data?

**Response:** Thank you for pointing this out. We are sorry for causing this confusion. The meaning of the sentence is the second option you described. It means gauged data were combined with remote sensing products and reanalysis data. We meant to use the phrase "a fusion of something. into *in-situ* station data" to express that the *in-situ* station observations are the backbone of the CMFD as He et al. (2020) mentioned in their paper. We have revised the sentence for better understanding: "Through a fusion of remote sensing products, reanalysis datasets, and *in-situ* station data, the China Meteorological Forcing Dataset (CMFD) has been produced to serve as a high-resolution (three hours, 0.1° × 0.1°) input forcing dataset for hydrological and ecosystem models beginning in 1979 (He et al., 2020)."

Section 2.3 and 2.4: Which method did the Authors use to re-grid the data? Where the raw data can be found?

**Response:** We used bilinear interpolation to regrid the data. We have added relevant descriptions of the regridding method into two sections as follows:

(In section 2.3) "We resampled the SRTM-DEM into 0.05° × 0.05° grid cells using the bilinear interpolation method."

(In Section 2.4) "The original spatial resolution is 0.04° × 0.04° for the monthly climatology of PRISM between 1961 and 1990; we used bilinear interpolation to regrid the spatial resolution into 0.05° × 0.05° grid cells for adjustment based on climatology."

The raw SRTM-DEM data can be found at this link: https://cmr.earthdata.nasa.gov/search/concepts/C1214622194-SCIOPS. We have updated the link in the main text. And the raw monthly climatology from PRISM can be found here: https://prism.oregonstate.edu/

Line 173: is it possible to eliminate interpolation errors?

**Response:** Thanks for your question. The interpolation errors cannot be eliminated entirely but

just reduced as much as possible. The word "eliminate" we used here is inappropriate and is misleading. We have revised the expression as follows: "To avoid this and reduce introduced errors, the overall strategy for establishing a daily gridded precipitation dataset is to construct a relatively continuous daily climatology field (Shen et al., 2010)."

Line 200: in the 30-year mean daily precipitation, was there any trend in the data or inhomogeneity?
**Response:** Thanks for the question. We calculated the trend in the 30-year mean daily precipitation. About two-thirds of stations have no significant trend for the 30-year mean daily precipitation.

As for inhomogeneity, we have previously tested the homogeneity of the gauge-based raw monthly precipitation series using the software package RHtestsV4 (Wang, published online July 2013). RHtestsV4 recommends testing the monthly series first before testing the corresponding daily series because daily series are much noisier and thus more difficult to test for changepoints. In RHtestsV4, two types of changepoint are detected: 1) Type-1 changepoints, which can be detected as significant at the nominal level even without metadata support (and if there is no significant changepoint identified, the time series being tested can be declared to be homogeneous); and 2) Type-0 changepoints, which can be significant only if they are supported by reliable metadata. In this study, we test for the existence of a Type-1 changepoint. Results show that, out of all 2,839 gauges, the monthly precipitation series between 1961 and 2022 is homogeneous for 2,133 gauges. Therefore, we ignore the impact of inhomogeneity in this version of dataset.

Lines 245 - 325: Four interpolation methods to construct the field of ratio. Still missing how they differ, why those have been chosen, and why they provide different results.
**Response:** Thanks for the comments. The four interpolation methods (angular-distance weighting (ADW), inverse distance weighting (IDW), thin plate spline (TPS), and triangulation-based nearest neighbor interpolation (TNNI)) for the field of ratio were selected based on three main principles: 1) Popularity—These four interpolation methods are widely used in generating daily gridded precipitation for various disciplines, such as atmospheric sciences (Ahrens, 2006), hydrological modelling (Ly et al., 2013), environmental management (Li and Heap, 2011), and civil engineering (Zhou et al., 2007). 2) Authority—Internationally, most of the currently prevailing meteorological datasets adopt one of these four interpolation methods. For example, the Climatic Research Unit

gridded Time Series (CRU TS) is developed using TNNI (Harris et al., 2014); Global land-surface precipitation data products of the Global Precipitation Climatology Centre (GPCC) are built based on ADW (Becker et al., 2013); and the China Meteorological Forcing Dataset (CMFD) is constructed using TPS (He et al., 2020). 3) Simplicity—Previous studies have demonstrated that the IDW method is a simple but efficient interpolation method (Ahrens, 2006). Statistical interpolation methods such as multiple linear regression, optimal interpolation, or kriging can perform better, but only if data density is sufficient (Eischeid et al., 2000).

Interpolation methods or interpolation functions are expected to be "smooth" (continuous and once differentiable), to produce values that will pass through the specified points (e.g., gauges), and to meet the user's intuitive expectations about the phenomenon under investigation (Shepard, 1968). Hence, there is a trade-off between local data fidelity and global fitting smoothness. To find the most appropriate interpolation method, we selected the four alternative interpolation methods to consider the balance between local data fidelity and global fitting smoothness in addition to the popularity, authority, and simplicity of each interpolation method. The ADW and IDW methods were chosen due to their high local data fidelity. Both are local interpolation methods (Liszka, 1984). However, unlike the IDW method, the ADW method assigns a tiny weight to far-distant gauge observations to promote global fitting smoothness. The TPS and TNNI methods, on the other hand, were chosen for their high global fitting smoothness. Both TPS and TNNI are global interpolation methods (Liszka, 1984). The TPS method is based on a mathematical model for surface estimation that fits a minimum-curvature surface through all input points, while TNNI constructs a Delaunay triangulation of three station locations. So TNNI tends to assign more weights to maintain local data fidelity.

Section 4.1 (starting line 369). My suggestion is to revise the term "optimal" for a scheme since there is no optimal scheme but simply the scheme having better metrics compared to the other schemes tested. The question of why such a combination of methods leads to better goodness of fit metrics is not answered. Why is such a combination better compared to the others? It is simply chance? Can this combination be transferred to other regions? Since the schemes perform differently depending on the topography (369-372), how do these differences affect the overall performance of the scheme? Are the metrics' values listed (lines 375-380) average over the 45k

stations used for verification?

**Response:**

Many thanks for your constructive suggestions. We have revised the phrase "optimal interpolation scheme" to "best-performing interpolation scheme among the selected metrics" in the manuscript.

We have discussed the main reasons for the differences among various gauge-based precipitation datasets in section 4.2. These reasons include the density of gauges involved and whether or not the interpolation scheme fully considers the impact of topography and boundary effects. Considering the spatiotemporal consistency of the daily gauge observations, the differences in interpolation performance among the eight interpolation schemes is mainly driven by whether the interpolation scheme fully considers the impact of topography and boundary effects.

The overall interpolation strategy was to combine the daily climatology field ($Cd$) with the field of the ratio between daily precipitation and daily climatology ($P/Cd$). With respect to $Cd$, the PRISM-type daily climatology field incorporates topographic features, proximity to coastlines, and several measures of terrain complexity, which goes beyond the climate–elevation relationships that the ANUSPLIN-type daily climatology field considers. As for $P/Cd,$ we selected the four alternative interpolation methods (angular-distance weighting (ADW), inverse distance weighting (IDW), thin plate spline (TPS), and triangulation-based nearest neighbor interpolation (TNNI)) to consider the balance between local data fidelity and global fitting smoothness in addition to the popularity, authority and simplicity of the interpolation methods. Specifically, the ADW and IDW methods were chosen due to their high local data fidelity. Both are local interpolation methods (Liszka, 1984). Unlike the IDW method, the ADW method assigns a tiny weight to far-distant gauge observations to promote global fitting smoothness. This impacts the local accuracy of interpolation. The TPS and TNNI methods, on the other hand, were chosen for their high global fitting smoothness. Both TPS and TNNI are global interpolation methods (Liszka, 1984). The TPS method is based on a mathematical model for surface estimation that fits a minimum-curvature surface through all input points, while TNNI constructs a Delaunay triangulation of three station locations. So TNNI tends to assign more weights to maintain local data fidelity but has weaker fitting smoothness. To sum up, the combination of PRISM-type $Cd$ and IDW-type $P/Cd$ yielded the best performance among the selected schemes. This was not simply due to chance. This bestperforming interpolation scheme could be applied in other regions, but further validation would be needed to confirm whether it is the best-performing interpolation scheme there.

And for the metrics' value in Lines 375-380, no, the metrics' values here are the median values over the ~45,000 stations used for verification.

Lines 479: Optimal interpolation scheme. Again, there is no optimal but the best among the ones tested

**Response:** Thanks for the suggestion. We have revised the term "Optimal interpolation scheme" in this sentence as follows:

"The median CC, RMSE, and KGE values for the interpolation scheme that performed the best among the selected metrics (in comparison with the high-density gauge observations used for validation) were 0.78, 8.8 mm/d, and 0.69, respectively."

References:

Ahrens, B.: Distance in spatial interpolation of daily rain gauge data, Hydrology and Earth System Sciences, 10, 197-208, https://doi.org/10.5194/hess-10-197-2006, 2006.

Becker, A., Finger, P., Meyer-Christoffer, A., Rudolf, B., Schamm, K., Schneider, U., and Ziese, M.: A description of the global land-surface precipitation data products of the Global Precipitation Climatology Centre with sample applications including centennial (trend) analysis from 1901–present, Earth System Science Data, 5, 71-99, https://essd.copernicus.org/articles/5/71/2013/, 2013.

Eischeid, J. K., Pasteris, P. A., Diaz, H. F., Plantico, M. S., and Lott, N. J.: Creating a Serially Complete, National Daily Time Series of Temperature and Precipitation for the Western United States, Journal of Applied Meteorology, 39, 1580-1591, https://doi.org/10.1175/1520-0450(2000)039<1580:CASCND>2.0.CO;2, 2000.

Harris, I., Jones, P. D., Osborn, T. J., and Lister, D. H.: Updated high-resolution grids of monthly climatic observations – the CRU TS3.10 Dataset, International Journal of Climatology, 34, 623-642, https://doi.org/10.1002/joc.3711, 2014.

He, J., Yang, K., Tang, W., Lu, H., Qin, J., Chen, Y., and Li, X.: The first high-resolution meteorological forcing dataset for land process studies over China, Scientific Data, 7, 25,

https://doi.org/10.1038/s41597-020-0369-y, 2020.

Li, J. and Heap, A. D.: A review of comparative studies of spatial interpolation methods in environmental sciences: Performance and impact factors, Ecological Informatics, 6, 228-241, http://www.sciencedirect.com/science/article/pii/S1574954110001147, 2011.

Liszka, T.: An interpolation method for an irregular net of nodes, International Journal for Numerical Methods in Engineering, 20, 1599-1612, https://doi.org/10.1002/nme.1620200905, 1984.

Ly, S., Charles, C., and Degré, A.: Different methods for spatial interpolation of rainfall data for operational hydrology and hydrological modeling at watershed scale: a review, Biotechnologie, Agronomie, Societe et Environnement, 17, 392-406, https://doi.org/10.6084/M9.FIGSHARE.1225842.V1, 2013.

Shepard, D.: A two-dimensional interpolation function for irregularly-spaced data, Proceedings of the 1968 23rd ACM national conference, 517–524, https://doi.org/10.1145/800186.810616, 1968.

Wang, X. L., Y. Feng: RHtestsV4 User Manual, Climate Research Division, Atmospheric Science and Technology Directorate, Science and Technology Branch, Environment Canada. 28 pp. [Available online at http://etccdi.pacificclimate.org/software.shtml]", published online July 2013.

Zhou, F., Guo, H.-C., Ho, Y.-S., and Wu, C.-Z.: Scientometric analysis of geostatistics using multivariate methods, Scientometrics, 73, 265-279, https://akjournals.com/view/journals/11192/73/3/article-p265.xml, 2007.